# Representation Consistency for Accurate and Coherent LLM Answer Aggregation

**Junqi Jiang**[1]    **Tom Bewley**[2]    **Salim I. Amoukou**[2]
**Francesco Leofante**[1]    **Antonio Rago**[3]    **Saumitra Mishra**[2]    **Francesca Toni**[1]
[1] Imperial College London    [2] J.P. Morgan AI Research    [3] King's College London
{junqi.jiang,f.leofante,f.toni}@imperial.ac.uk
{firstname.surname}@jpmorgan.com, antonio.rago@kcl.ac.uk

## Abstract

Test-time scaling improves large language models' (LLMs) performance by allocating more compute budget during inference. To achieve this, existing methods often require intricate modifications to prompting and sampling strategies. In this work, we introduce representation consistency (RC), a test-time scaling method for aggregating answers drawn from multiple candidate responses of an LLM regardless of how they were generated, including variations in prompt phrasing and sampling strategy. RC enhances answer aggregation by not only considering the number of occurrences of each answer in the candidate response set, but also the consistency of the model's internal activations while generating the set of responses leading to each answer. These activations can be either dense (raw model activations) or sparse (encoded via pretrained sparse autoencoders). Our rationale is that if the model's representations of multiple responses converging on the same answer are highly variable, this answer is more likely to be the result of incoherent reasoning and should be down-weighted during aggregation. Importantly, our method only uses cached activations and lightweight similarity computations and requires no additional model queries. Through experiments with four open-source LLMs and four reasoning datasets, we validate the effectiveness of RC for improving task performance during inference, with consistent accuracy improvements (up to 4%) over strong test-time scaling baselines. We also show that consistency in the sparse activation signals aligns well with the common notion of coherent reasoning.

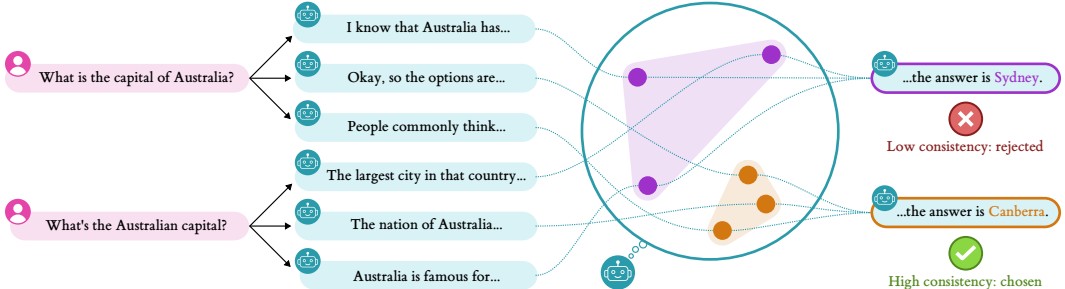

Figure 1: An illustrative example of representation consistency. When aggregating an answer from multiple LLM responses (in the blue boxes on the middle left) sampled from semantically equivalent rephrasings of the same question (in the pink box on the left), we take into account the consistency of model internal activations within each response group (points in the blue circle). In this case, the answer "Canberra" is chosen (on the right) because the activations of its corresponding responses are more similar (the area covered by the orange points is smaller than that of the violet points).

39th Conference on Neural Information Processing Systems (NeurIPS 2025).

# 1 Introduction

Deep learning models have achieved remarkable capabilities across tasks and modalities, with large language models (LLMs) [67, 9, 54] being perhaps the single most impactful model class [77]. As the performance gains from LLM pre-training [36] begin to slow down on many benchmarks, recent research has diverted focus (and compute) to boosting model performance at the time of inference [76]. This line of work is often referred to as test-time scaling, and has proven to be effective, especially as larger compute budgets become feasible [8].

Numerous test-time approaches have been proposed to improve the outputs of LLMs without needing to update the models themselves [71, 68, 37]. These approaches can be broadly divided into two categories [76]: prompt-based and decoding-based. The former usually requires more outputs from relevant models. This includes asking the same LLM to generate more complex outputs to elicit more reasoning [71, 47], obtaining more responses (either through sampling [68] or prompt phrasing [60]), then aggregating answers, and incorporating auxilliary LLMs to collectively decide on outputs [19, 37]. This can be done in a single-turn or multi-turn conversation format. A simple yet effective approach is self-consistency (SC) [68], which takes the modal answer from multiple diversely sampled chain-of-thought (CoT) [71] responses. On the other hand, decoding-based methods intervene on the regular decoding process to obtain higher-quality responses [73, 5, 52].

However, methods like SC ignore an important source of information: the internal model activations. These activations are known to contain rich encodings of LLMs' working mechanisms [16, 4], the study of which may help us better understand how they function, such as identifying neurons related to factual knowledge storage [74, 13] and signals useful for uncertainty quantification [11]. While recent works investigate feeding the activations back into the response generation loop for reasoning improvements in single LLM responses [24, 59, 38, 56], they require model retraining. The usefulness of model activations cached in multiple LLM responses for test-time scaling has not been studied.

We fill in this gap by proposing representation consistency (RC), a method to enhance LLM answer aggregation from multiple responses using internal activations. We focus on the setting where, given a question, multiple prompt rephrasings can be generated for that question, and multiple LLM responses can be sampled for each rephrasing (Section 2). This is inspired by recent works showing that answers obtained in such scenarios are more robust and better reflect LLM uncertainty [57, 15]. Given a set of responses generated in this way, RC groups the responses by the final answer they give to the question, and uses an evaluation function to score each response group. This function combines the consistency (measured by a similarity metric) among the activations of each response in the group, and the cardinality of the group relative to the total number of responses generated (Section 3). We propose two variations of RC using both dense raw model activations and sparsified signals via sparse auto-encoders (SAEs) [30], specialised neural networks that can extract disentangled concepts from activations. Our intuition is that if multiple responses leading to the same answer have wildly differing activations, the model's reasons for selecting that answer could be inconsistent, so this answer is less likely to be correct. Conversely, greater similarity in the activations of multiple responses (which may be diverse in their surface-level content) implies a more robust underlying reasoning process, which may be indicative of correctness. Importantly, RC does not require specialised prompting or modifications to the decoding procedure. Instead, it only uses the cached activations from generation and lightweight similarity computations. Figure 1 depicts an illustrative example.

We evaluate RC on four popular open-source LLMs of varying sizes (2B to 27B) and on diverse reasoning datasets (Section 4). We show that by using model internals, there are consistent accuracy improvements (up to 4% in some settings) over SC, and over another strong baseline that we adapt using the same intuitions as RC but with external embedding models [18, 12]. We also provide some experimental insights showing the consistency in the sparsified activation space aligns well with the common notion of coherent reasoning among a group of responses.

# 2 Problem Definition

We focus on single-turn question-answering (QA) tasks for which ground truth answers exist, and consider a setting involving multiple prompt rephrasings and sampled LLM responses.

Formally, let $\mathcal{S} = \bigcup_{l=1}^{\infty} \mathcal{V}^l$ be the set of all possible token sequences constructed from a vocabulary $\mathcal{V}$. An LLM $\pi : \mathcal{S} \rightarrow \Delta(\mathcal{S})$ is a function that takes in a *prompt* sequence $x \in \mathcal{S}$ and outputs a distribution over *response* sequences $y \in \mathcal{S}$, from which individual responses can be sampled.

A QA task is a tuple $\tau = (\mathcal{Q}, \mathcal{A}, \rho, phrase, parse, \ell)$. $\mathcal{Q}$ is a set of questions and $\mathcal{A}$ is a set of answers. $\rho \in \Delta(\mathcal{Q} \times \mathcal{A})$ is a distribution from which questions $q$ and ground truth answers $a^*$ are sampled. $phrase : \mathcal{Q} \rightarrow \Delta(\mathcal{S})$ is a stochastic function that maps a question $q$ to a distribution over prompt sequences $x$ (concretely: alternative rephrasings of the question), and $parse : \mathcal{S} \rightarrow \mathcal{A}$ is a deterministic function (e.g. regular expression matching) that maps a response sequence $y$ to an answer $a$.[1] $\ell : \mathcal{A} \times \mathcal{A} \rightarrow \mathbb{R}_{\geq 0}$ is a non-negative loss function between the answer $a$ and the ground truth $a^*$ (lower is better). $\ell$ is readily linked to performance metrics like accuracy or ROUGE score.

We now describe two baseline methods for this problem setting.

**Non-selective Expectation (NE)** refers to the expected loss of an LLM $\pi$ over each response from each question rephrase, which can be quantified as follows:

$$\mathcal{L}_{NE}(\pi, \tau) = \mathbb{E}_{(q,a^*) \sim \rho} \, \mathbb{E}_{x \sim phrase(q)} \, \mathbb{E}_{y \sim \pi(x)} \, \ell(parse(y), a^*). \tag{1}$$

**Self-consistency (SC)** [68] is a strong baseline for this problem setting. For a given question $q$, it takes the most common (modal) answer from a finite sample of prompts and responses $\mathcal{D}_q = \{(x_i \sim phrase(q), y_i \sim \pi(x_i))\}_{i=1}^{N}$:

$$modal(\pi, q) = \arg\max_{a \in \mathcal{A}} \big| \{(x_i, y_i) \in \mathcal{D}_q : parse(y_i) = a\} \big|. \tag{2}$$

This method leverages the intuition that correct answers are likely to emerge more often among a sufficiently large and diverse set of reasoning paths. The expected loss of this method is:

$$\mathcal{L}_{SC}(\pi, \tau) = \mathbb{E}_{(q,a^*) \sim \rho} \, \ell(modal(\pi, q), a^*). \tag{3}$$

We note that when there are multiple modal answers, there is no clear tie-breaking mechanism apart from randomly choosing an answer.

Our goal in this work is to develop an alternative method for selecting between responses from $phrase$ and $\pi$ to reduce the expected task loss beyond that achieved by NE and SC.

## 3 Representation Consistency

Next, we introduce representation consistency (RC) for answer aggregation, leveraging an additional source of information: the internal activations of the LLM $\pi$ as it generates each response. Since these activations encode rich information about the LLM's reasoning processes, our intuition is that if multiple responses leading to the same answer have distinct activations, then the model's reasons for selecting that answer may be inconsistent, suggesting the answer is more likely to be incorrect. We show this consideration can be combined with each answer's frequency within the response set to create a criterion for aggregation.

Let $f_\pi : \mathcal{S} \times \mathcal{S} \rightarrow \mathcal{Z}_\pi$ be a function that maps a pair of prompt and response sequences to the internal activation space of a language model $\pi$, such that $f_\pi(x, y)$ denotes $\pi$'s activation when generating response $y$ to prompt $x$. For each question $q$, activations for $\mathcal{D}_q$ grouped by answer $a$ can be cached:

$$Z_{q,a} = \{f_\pi(x_i, y_i) : (x_i, y_i) \in \mathcal{D}_q, \, parse(y_i) = a\}. \tag{4}$$

Note that for an autoregressive transformer model, it is common to take this to be the residual stream activation at a particular layer (usually around the middle), either at one target token position or across all token positions [1]. Given all model activations for one generation process (over a prompt-response pair) $z \in \mathcal{Z}_\pi$, we denote its 1-d slice at layer $l$ and token position $n$ as $z_n^l \in \mathbb{R}^{d_\pi}$ where $d_\pi$ is the LLM's hidden dimension. Then, we introduce an evaluation function $V : \bigcup_{n=0}^{\infty} \mathcal{Z}_\pi^n \rightarrow \mathbb{R}$ to score each answer considering both their supporting responses' internal activations and their frequency among all responses. Specifically, the evaluation function is a sum of two components weighted by a hyperparameter $\lambda \in [0, 1]$:

$$V_{q,a} = \lambda \cdot consistency_{q,a} + (1 - \lambda) \cdot frequency_{q,a}. \tag{5}$$

---

[1]This parsing notation is very general. For instance, it can handle cases where the response contains chain-of-thought reasoning, in which case only the final answer is parsed. It also handles cases where the response fails or refuses to answer the question, provided the answer set $\mathcal{A}$ contains a corresponding 'null' answer.

**Consistency** measures the representational similarity between the activations (given some layer $l$ and token position $n$) of each response within a response group:

$$consistency_{q,a} = \frac{1}{|Z_{q,a}|(|Z_{q,a}| - 1)} \sum_{z_1 \in Z_{q,a}} \sum_{z_2 \in Z_{q,a} \setminus \{z_1\}} sim(z_{1,n}^l, z_{2,n}^l), \tag{6}$$

where $sim : \mathbb{R}^{d_\pi} \times \mathbb{R}^{d_\pi} \to [0, 1]$ is a similarity metric. This notation is generic, and we use cosine similarity in this paper.

**Frequency** measures the proportion of responses supporting each answer in $\mathcal{D}_q$:

$$frequency_{q,a} = \frac{|\{(x_i, y_i) \in \mathcal{D}_q : parse(y_i) = a\}|}{|\mathcal{D}_q|}. \tag{7}$$

Then, RC selects an aggregated answer as:

$$RC(\pi, q) = \arg \max_{a \in \mathcal{A}} V(Z_{q,a}), \tag{8}$$

and the expected loss is:

$$\mathcal{L}_{RC}(\pi, \tau) = \mathbb{E}_{(q,a^*) \sim \rho} \, \ell(RC(\pi, q), a^*). \tag{9}$$

In RC, consistency in activations and the frequency of each answer are considered jointly. This handles the fact that groups of responses per answer may vary in size, including the possibility of some having only one or zero answers. When two answer groups are similar in size, the evaluation function $V$ mostly decides on the final answer via their consistency. Conversely, it tends to take the modal answer if this answer's group is much larger than the rest, unless its consistency is very low. The trade-off is controlled by $\lambda$. We note that when $\lambda = 0$, the evaluation function reduces to taking the modal answer, so RC becomes identical to SC. When $\lambda = 1$, the selection is based purely on activation similarity. This can be undesirable because any answer group with only one answer will always be chosen. As shown in our experiments (Section 4), the $\lambda$ hyperparameter needs to be tuned for optimal performance and is model- and dataset-specific. Additionally, the model layer $l$ from which the activations are taken is also tunable, although a layer near the middle of the model architecture often gives the best results.

**The sparse variation** Raw model activations are known to be densely polysemantic [1], and human-understandable information can be encoded across model layers [4]. This means that calculating similarity on them might include redundant, noisy information. To counter this potential issue, we propose an extra step for RC with Sparse-AutoEncoders (SAEs). They are neural network models operating on LLM activations to encode them into sparse and well-disentangled signals in SAEs' latent space [30]. Each layer of any LLM can be associated with an SAE trained on its activations. We denote the encoder of the SAE for the $l$-th layer as $f_{enc}^l : \mathbb{R}^{d_\pi} \to \mathbb{R}^{d_{SAE}}$, and $d_{SAE}$ is the number of latent dimensions of the SAE. Then, the only change from RC is the way consistency is calculated, with $sim$ function handling different input sizes:

$$consistency\text{-}sparse_{q,a} = \frac{1}{|Z_{q,a}|(|Z_{q,a}| - 1)} \sum_{z_1 \in Z_{q,a}} \sum_{z_2 \in Z_{q,a} \setminus \{z_1\}} sim(f_{enc}^l(z_{1,n}^l), f_{enc}^l(z_{2,n}^l)). \tag{10}$$

Incorporating SAEs only introduces slight additional computational costs to the standard RC. Apart from the memory space needed for caching the LLM activations, when $l$ and $\lambda$ are determined, the additional cost for an input text is one forward pass in an SAE. Specifically, the input to SAEs, an LLM's residual activation at one token position, is of shape $(1, d_\pi)$, and the number of parameters in each SAE is usually small (e.g., a pretrained SAE for the Gemma-2-9B model [65] has 0.1B parameters [44]). These indicate that the additional costs by SAEs are only a fraction of an LLM's forward pass over a prompt.

We respectively refer to the RC instantiations with raw model activations and with sparsified activations as **RC-D** (dense) and **RC-S** (sparse).

# 4 Experiments

In this section, we quantitatively evaluate the effectiveness of RC for improving LLMs' task performance at test time. We first introduce our experiment setup.

**Models**   We experiment with 4 open-source LLMs of varying sizes, `Llama-3.1-8B-Instruct` [21], `Gemma-2-2B-IT`, `Gemma-2-9B-IT`, `Gemma-2-27B-IT` [65]. These models are chosen because our method applies to open-source LLMs whose internal activations are obtainable. Also, researchers have pretrained open-source SAEs for these models, namely GemmaScope [44] and LlamaScope [25], which we use to investigate the differences between RC with raw activations and with sparsified activation signals. See Appendix A.1 for details on how they are used.

**Datasets**   We test all methods using four reasoning datasets spanning diverse topics, CommonsenseQA (CSQA) [62] for common sense reasoning, MMLU [26] for exam-style questions, MedMCQA [55] for medical domain-specific knowledge, and Hellaswag (HSwag) [75] for sentence completion tasks. We experiment with multiple-choice QA datasets because of the need to efficiently group generations by their final answer. We use the test or eval sets of these datasets - 1200 samples for CSQA, and 3000 samples for other datasets. For `Gemma-2-27B-IT`, we experiment with 1000 samples per dataset due to the heavy computation load.

**Obtaining multiple LLM responses**   Each response is generated with CoT prompting, i.e., first ask for some step-by-step analysis, then output a final choice [71]. All prompts are zero-shot. We use the following configurations (number of responses = number of prompt phrasings × number of samples from each prompt): 12 responses with $(12 \times 1)$, $(6 \times 2)$, $(4 \times 3)$, $(3 \times 4)$, $(2 \times 6)$, $(1 \times 12)$, and 6 responses with $(6 \times 1)$, $(3 \times 2)$, $(2 \times 3)$, $(1 \times 6)$. We chose 12 and 6 because they cover a large range of combinations for prompt-sample numbers. Every rephrased prompt is semantically equivalent. The ways they present the question and the candidate choices in the dataset samples are identical, and they only differ in the CoT instructions. For example, "*Provide short explanations of your thinking steps*", and "*Briefly justify your reasoning process*". See Appendix B for more details on prompts. Model generations from each prompt are sampled with 0.7 temperature for balanced randomness. Regular expression matching is used to extract the chosen answer from each generation for multiple-choice questions.

**Baselines**   In addition to **NE** and **SC** (Section 2), we incorporate another baseline adapted from the literature. It can be regarded as a variation of RC, which we refer to as **RC-E** (external). RC-E is in the same spirit as RC, but instead of using the LLM's internal activations to calculate consistency (Equation 6) for the evaluation function (Equation 5), RC-E uses embeddings (of the LLM-generated responses) from external encoder language models for this purpose. This way, we provide an ablation for examining the effectiveness of incorporating internal activations against using embeddings from external sources. We introduce RC-E below.

Let $f_{nli} : S \times S \to \mathbb{R}^3$ be a natural language inference (NLI) model. Given two sentences $y_1, y_2 \in S$, $f_{nli}(y_1, y_2) = [p_{entail}, p_{neutral}, p_{contradict}]$ predicts the probabilities that the semantic meaning of $y_1$ entails, is neutral to, or contradicts that of $y_2$, where $p_{entail} + p_{neutral} + p_{contradict} = 1$. We denote the first output element, entailment probability, as $f_{nli}^{ent}(y_1, y_2)$.

RC-E proceeds as follows. Given a finite sample of prompts and responses for each question $q$, $\mathcal{D}_q = \{(x_i \sim phrase(q), y_i \sim \pi(x_i))\}_{i=1}^N$, first calculate average entailment probabilities among each group of generations $\mathcal{Y}_{q,a} = \{y_i | (x_i, y_i) \in \mathcal{D}_q, parse(y_i) = a\}$:

$$EP_{q,a} = \frac{1}{|\mathcal{Y}_{q,a}|(|\mathcal{Y}_{q,a}| - 1)} \sum_{y_1 \in \mathcal{Y}_{q,a}} \sum_{y_2 \in \mathcal{Y}_{q,a} \setminus \{y_1\}} f_{nli}^{ent}(y_1, y_2) \tag{11}$$

Then, the answer is selected by using a similar evaluation function to RC with a hyperparameter $\lambda$:

$$RC_{external}(\pi, q) = \arg\max_{a \in \mathcal{A}} \ (\lambda \cdot EP_{q,a} + (1 - \lambda) \cdot frequency_{q,a}) \tag{12}$$

where $frequency$ is the same as that in RC (Equation 7). This method is inspired by how NLI (and embedding) models are used in uncertainty quantification [39, 15] and retrieval-augmented generations [43]. In our experiments, we use `bge-m3-zeroshot-v2.0` [12, 41] for NLI with long sequence lengths. For all baselines, when there is a tie during answer aggregation, we select the last answer for reproducibility.

**RC instantiations**   We use both the raw model activations and the SAE-encoded activations to instantiate RC, respectively **RC-D** (dense) and **RC-S** (sparse). We follow the common practice of taking activations from the residual stream as in the mechanistic interpretability literature [1]. For `Gemma-2-9B-IT` and `Gemma-2-27B-IT` models, we retrieve activations from the three layers on which GemmaScope SAEs are trained, located at 25%, 50%, 75% of model depth. For the other two models, their pretrained SAEs cover every layer, and we take activations from layers at 10%, 25%, 50%, 75%, and 90% of model depth. For each LLM generation, we cache the activation at the token location where the model is about to output the final answer choice, e.g., right after "the answer is: ", and before "A". In terms of the $sim$ function in Equations 6 and 10, we use cosine similarity to capture directions in the latent space. See Appendix A.2 for a detailed discussion on the implementation.

We then perform two sets of evaluations. In Section 4.1 we report the task performance (accuracy) of each method, and in Section 4.2 we investigate whether the consistency in the representation space of LLM aligns with the common notion of coherent reasoning. All experiments are executed on a Linux machine with 3 NVIDIA A100 GPUs, each with 80GB memory.

## 4.1   Task Performance Results

Throughout this section, we use SC as the main baseline and report every other method's accuracy relative to SC. We use 50% data to find the optimal hyperparameters for each method ($\lambda$ for RC-E, $\lambda$ and model layer for RC), and report task performance results on the remaining 50%. We focus on the test samples where there exist multiple answers from the set of LLM responses (see Appendix C.1 for more detail) to better distinguish the performance differences between SC, RC-E and RC, as their chosen answer would be the same for the remaining samples.

The task performances averaged for each model are summarised in Table 1. Figure 2 further reports detailed results for each model and dataset, averaged over prompt-sample configurations for 12 and 6 answers, respectively. Note that we include 4 models, 4 datasets, and 10 different prompt-sample configurations, making in total 160 sets of experiments. See Appendix C.2 for full results.

Table 1: Accuracy results (%) summarised for each model. We report the absolute results for the main baseline, SC, and relative results to SC for the remaining methods

|      | Llama3.1-8B-IT | Gemma2-2B-IT | Gemma2-9B-IT | Gemma2-27B-IT |
|------|----------------|--------------|--------------|---------------|
| NE   | -5.60          | -4.43        | -4.37        | -5.19         |
| SC   | 52.9           | 44.7         | 48.6         | 52.3          |
| RC-E | +1.06          | +0.84        | +1.07        | +0.55         |
| RC-D | **+1.84**      | +0.89        | +1.32        | +0.76         |
| RC-S | +1.73          | **+1.10**    | **+1.40**    | **+0.89**     |

From Table 1, it is clear that RC-D and RC-S show the highest average accuracy gains for each model. RC-E moderately improve over SC, which is already a strong baseline when compared with NE - the average CoT accuracy. Zooming in to detailed results for each number of responses configuration of each dataset (Figure 2), again, we observe that in most cases (30 out of 32 plots), the RC-D and RC-S yield the highest performance gains, often exceeding 1% and sometimes over 2%. RC-S more frequently outperforms RC-D. The best performance is observed at CSQA dataset for the Llama model at 6 responses, RC-D and RC-S respectively have 3.85% and 4.00% accuracy improvements, while RC-E is also effective here. However, there are rare cases where all RC variations show no improvements over SC, and we have observed decreased performance in individual cases (Appendix C.2). Overall, the improvements are more obvious on smaller models and with 6 responses, and are less so on the counterpart settings, possibly due to the stronger baseline performance.

Table 2 provides some insights into the optimal hyperparameters of RC-D and RC-S. We found that the most common optimal model layer for RC is at the middle, which coincides with recent research works stating that these layers often contain diverse high-level information [66, 50]. $\lambda$ values are model- and dataset-specific, especially when the number of test samples is limited. Overall, the optimal relative importance of consistency and frequency in RC (Equation 5) is close to 1:1. See Appendix D for a detailed ablation study.

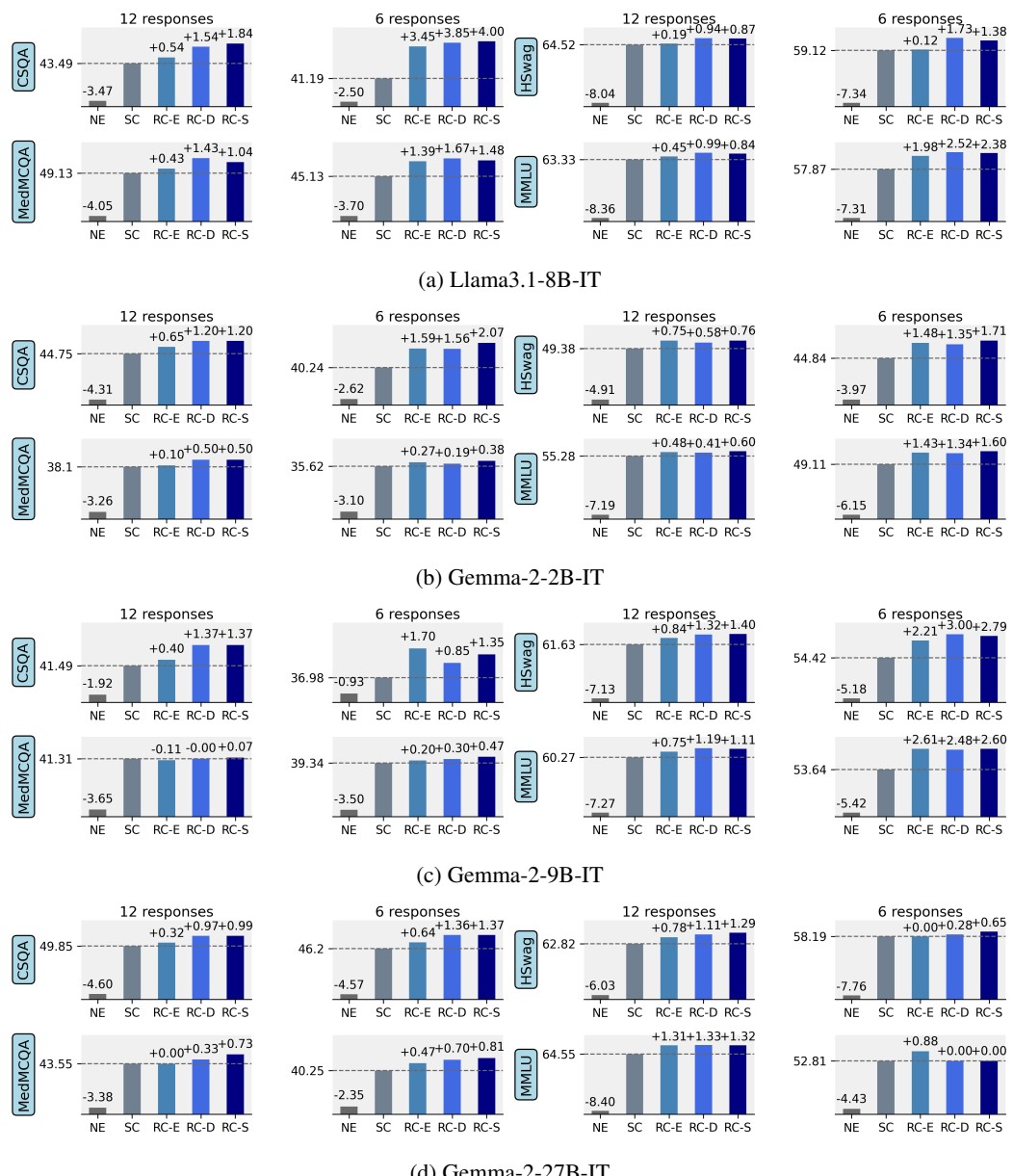

Figure 2: Accuracy results (%) summarised for each number of responses configuration, dataset, and model. We report the absolute results for the main baseline, SC to the left of each subfigure with a dashed line for easy comparison. We report the relative results to SC for the remaining methods, the performance difference are shown on top of each bar.

Table 2: Optimal hyperparameters of RC for each model. The first number is the depth percentage of the model layer where activations are taken, the second number is the averaged optimal $\lambda$ value

|  | Llama3.1-8B-IT | Gemma2-2B-IT | Gemma2-9B-IT | Gemma2-27B-IT |
|---|---|---|---|---|
| RC-D | 50%, 0.43 | 25%, 0.29 | 50%, 0.36 | 50%, 0.44 |
| RC-S | 25%, 0.73 | 50%, 0.45 | 50%, 0.46 | 50%, 0.59 |

We additionally perform an analysis validating the non-trivial usefulness of LLMs' internal activations for aiding answer selection, compared against the surface-level textual outputs. For each model, we

Table 3: Percentages where the correct/incorrect answer is associated with a higher consistency, as calculated by each method. The number after each model name is the number of interested cases matching the description.

| Method/Model | Gemma-2-2B-IT (497) | Gemma-2-9B-IT (404) | Gemma-2-27B-IT (113) | Llama-3.1-8B-IT (195) |
|---|---|---|---|---|
| Embedding (baseline) | 50.9%/49.1% | 49.5%/50.5% | 47.8%/52.2% | 50.3%/49.7% |
| Dense (ours) | 53.3%/46.7% | 53.7%/46.3% | 57.5%/42.5% | 56.4%/43.6% |
| Sparse (ours) | 53.9%/46.1% | 54.7%/45.3% | 58.4%/41.6% | 56.4%/43.6% |

collected the test cases where among the 12 responses, 6 support one answer and the other 6 support another answer, and one of these two answers is the correct one. We count the number of times the correct vs. incorrect answer is associated with a higher consistency (dense activation consistency (ours, Equation 6), sparsified activation consistency (ours, Equation 10), and embedding consistency (the entailment probability baseline, Equation 11)). We aggregate the results for each model over all prompt-sample configurations and over the datasets. This is presented in Table 3. Note that the consistency measures are prone to the cases where the LLM is confidently wrong about some answer, so the absolute values of such measures could be tightly linked to the LLM's performance. We observe that the correct answer constantly demonstrates a higher consistency calculated with internal activations, while this is not the case with textual embeddings. This could indicate that the activations help reveal information during the response generation process that is not included in the textual outputs.

## 4.2 Answer Coherence Results

Table 4: Percentage of times (%) where a higher internal representation consistency (RC-D, RC-S) or entailment probability in RC-E corresponds to more coherent reasoning (labelled by deepseek-R1) among a group of responses.

| Dataset | Method | Llama3.1-8B-IT | Gemma2-2B-IT | Gemma2-9B-IT | Gemma2-27B-IT |
|---|---|---|---|---|---|
| CSQA | RC-E | 51.0 | 55.9 | 59.6 | 62.5 |
| | RC-D | 67.6 | 60.4 | 69.4 | 72.9 |
| | RC-S | **94.6** | **91.1** | **90.4** | **93.8** |
| HSwag | RC-E | 55.6 | 55.5 | 55.4 | 61.7 |
| | RC-D | 51.4 | 44.7 | 56.8 | 65.0 |
| | RC-S | **91.6** | **85.2** | **94.4** | **95.9** |
| MedMCQA | RC-E | 44.7 | 53.0 | 50.4 | 48.1 |
| | RC-D | 47.1 | 49.0 | 63.8 | 59.1 |
| | RC-S | **98.8** | **88.0** | **92.1** | **90.9** |
| MMLU | RC-E | 57.0 | 54.2 | 49.3 | 56.4 |
| | RC-D | 48.3 | 47.9 | 55.5 | 58.2 |
| | RC-S | **92.8** | **81.6** | **82.9** | **90.0** |

Next, we investigate the alignment between the consistency of model representations and what we, as humans, would perceive as coherent reasoning. We focus on the test cases (in each prompt-sampling configuration with 12 candidate responses) where the model produces 2 answers, and the number of responses supporting each answer is close, i.e., 6 vs. 6 and 5 vs. 7. To approximate our coherence notion, we employ LLM-as-a-judge, and discuss LLMs' suitability for this evaluation in Appendix E. Specifically, we prompt the deepseek reasoner model (deepseek-R1) [22], asking which group of responses is more coherent in their reasoning. Using these results, we then record the percentage of times when the group of responses with better consistency in their representations (Equation 6, we use the 50% depth layer for every model) coincides with the coherence label. We perform the same evaluation with RC-E's entailment probability (Equation 11) as a baseline.

The results are shown in Table 4. RC-S achieves the highest agreement rate with the labels, surpassing 90% in most cases. RC-E and RC-D have similar performance, near 50%, with RC-D more frequently having higher scores.

It is surprising to observe a substantial gap between RC-S and RC-D. This indicates that despite their similar task performances, they rely on different notions of consistency to achieve them. We can confirm that the model activations, after being processed by SAEs, align well with the common coherence notion. This is in line with the training objectives of SAEs - sparsify the dense signals in raw activations into sparse, potentially human-understandable concepts. We observe in our experiments that, among the many ($> 10k$) SAE latent dimensions, only about 100 are activated (having non-zero values) at the target token position during generation. Therefore, a larger cosine similarity here means similar concepts are present when determining the final answer. This intuition naturally matches our notion of coherence. Similarly, RC-E's entailment probability aims at obtaining the more coherent set, but is highly sensitive to the performance of the external NLI model on LLMs' CoT-style responses.

On the other hand, dense raw activations at single layers may carry more information than SAE activations. The accuracy improvements (Section 4.1) and the lower agreement rates (Table 4) hint at useful but less interpretable representational consistency notions different from what we understand as coherence.

## 5    Related Work

**Test-time scaling** refers to the problem of improving the task performance or output quality of LLMs after they have been trained [76]. It is distinct from post-training [9], which updates model parameters via supervised fine-tuning [2] or reinforcement learning [58], instead seeking to make improvements without directly modifying the model. The prominent approaches are related to chain-of-thought reasoning, asking for some intermediate reasoning process [71]. This originally works on each response in a single-turn conversation. Further research extends this to ask for self-reflection and correction [47] on the same LLM, forming multi-turn generations. There are also works to incorporate multiple LLMs performing similar conversations to reach more robust outputs [20], with merged vocabularies across models [29, 72], or in multiple turns [37, 19]. Works like self-consistency [68], instead, obtain multiple responses from a single model (possibly in multi-turn [64]) and then perform answer aggregation. Others bring a mix of the above approaches and intervene in the decoding process [73, 5, 52]. Different to these approaches, we consider model internal activations for aggregating multiple LLM responses.

**The role of multiple prompts**    It is known that outputs of language models are very sensitive to the prompts, even those semantically equivalent rewritings [57, 46, 14]. However, this setting has proven to be useful [45]. Similarly to SC, ensembling outputs over multiple prompts can improve task performance [35, 60]. When combined with diverse sampling, prompt rephrasing also help better quantify the uncertainty [28, 15] and improve calibration [34] in LLMs. Additionally, if viewing phrasing of the question as part of a predictive model, prompt sensitivity also resembles model multiplicity in machine learning [48, 7], which states the existence of multiple equally performing models that could give different predictions. Ensembling-based methods are often used to address this issue [6, 33]. Our method readily applies to multiple prompts and samples.

**Model internal activations** are at the core to mechanistic interpretability research (see [4] for a recent overview), which aims to understand model behaviours by investigating patterns in these activations. By training probe classifiers on the activations, researchers have identified neuron locations that correspond to actual knowledge [49, 74, 13], representations for uncertainty [11], and hallucination risks [31, 61]. Useful directions in the activation space have also been identified for steering LLMs towards desirable behaviours like instruction following [10, 63, 42]. Apart from these observational approaches, patching methods intervene on an LLM's forward pass by replacing activations at certain locations with those obtained from other runs to perform tasks like identifying critical neurons [51] and finding parts of the LLM for specified behaviours [23]. Finally, sparse autoencoders are intermediate tools trained on LLM activations to map them into disentangled concepts, such that only a small portion of SAE latent dimensions are activated during each generation [30, 1, 44, 25]. In our case, we use both raw activations and SAE-encoded sparse signals for test-time scaling.

# 6   Conclusions

In this work, we introduce representation consistency (RC), a method using model internal activations to enhance answer aggregation from multiple LLM responses. To the best of our knowledge, we are the first to investigate the usefulness of activations for such test-time scaling scenarios. We propose two variations of RC that respectively operate on the dense raw model activations and their SAE-sparsified counterparts. In our experiments, we also adapt existing works into a new ablation baseline that leverages the same intuitions behind RC but with external embedding models. We show that these new methods all consistently improve over the strong baseline, self-consistency, with RC-sparse delivering the highest accuracy gains. Additionally, we show that the representational consistency in the latent space of LLMs' corresponding SAEs aligns well with what we would deem as coherent reasoning in multiple responses.

Our proposed method comes with some limitations. First, RC requires access to model activations during generation. While developers of proprietary LLMs can make use of RC, other users could only apply RC to open-source models. Second, our intuitions behind RC may break in cases where an LLM is confidently and systematically wrong in its predictions, so that an answer resulting from multiple paths of incorrect reasoning may nonetheless have consistent activations. However, we suspect that any method that operates on given LLM responses without further model training would struggle to handle such cases. Also, our encouraging accuracy improvement results suggest that the heuristic of representation consistency is effective for ruling out many incorrect answers, and improving answer aggregation on average.

Exciting future works are envisaged following the introduction of RC. While we have investigated its integration with LLM response sets obtained by sampling and prompt rephrasing, the method could also be combined with more complex decoding-based scaling methods, e.g. tree-of-thoughts [73]. In this work, we examined RC on multiple-choice tasks. It would be an interesting direction to extend it to other reasoning tasks, possibly with more complex and open-ended generation forms. While designed for handling multiple responses from a single LLM, investigating RC's transferability across multiple LLMs would also be desirable [53]. Consistency from multiple responses has been a useful signal for model training [3, 17, 69], and using activations to aid training has been studied [24, 59]. It would be interesting to explore how the consistency of activations can help in improving reasoning or model controllability. There are recent works separately using model internals [11] or prompt rephrasings [15] for uncertainty quantification and hallucination mitigation. Our method, combining both elements, could potentially be studied in this space too. Finally, our findings on the mismatch between raw model activation consistency and our understanding of coherence also highlight the need for further studies on the interpretability and transparency of LLMs [27], possibly aided by SAEs.

## Disclaimer

## Acknowledgements

Jiang, Rago and Toni were partially funded by J.P. Morgan and by the Royal Academy of Engineering under the Research Chairs and Senior Research Fellowships scheme. Leofante was funded by Imperial College London through under the Imperial College Research Fellowship scheme. Rago and Toni were partially funded by the European Research Council (ERC) under the European Union's Horizon 2020 research and innovation programme (grant agreement No. 101020934). Any views or opinions expressed herein are solely those of the authors listed.

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

# A   Implementation Details

## A.1   Models

When obtaining responses from the experimented LLMs, we apply their respective chat templates described in their `huggingface` model card. We query them with the `transformers` library. To accelerate inference, responses from Gemma models are queried with a batch size of 6. No batching is applied to Llama model because it does not have a dedicated padding token, and is by default right-padded which conflicts with the implementation step to cache model activations.

We use the Python library `sae-lens` (`https://jbloomaus.github.io/SAELens/sae_table/`) for using the SAEs. Specific SAE models and the layers are summarised in Table 5. They can be found in the `sae-lens` link above. Note that SAEs for 2B, 27B Gemma models and the Llama model are originally trained on the activations of their base version. It has been shown that the SAEs also work on their instruction-tuned version [44, 25].

Table 5: SAE models used and the layer numbers at each LLM's respective model depth

| SAE name | 10% | 25% | 50% | 75% | 90% |
|---|---|---|---|---|---|
| llama_scope_lxr_8x | 3 | 8 | 16 | 24 | 29 |
| gemma-scope-2b-pt-res-canonical | 3 | 7 | 13 | 20 | 23 |
| gemma-scope-9b-it-res-canonical | - | 9 | 20 | 31 | - |
| gemma-scope-27b-pt-res-canonical | - | 10 | 22 | 34 | - |

## A.2   RC Implementation

Detailed implementations can be found in the accompanying code. We use off-the-shelf functionalities provided by `sae-lens` and perform activation caching in a two-step process for large-scale experiments. For each experiment on a model, a dataset, and for all prompt-sample configurations, we first generate all the answers needed using the `transformers` library. Then, we process the answer to identify the token location in the response where the model is about to output the final answer choice. In a separate process, we then concatenate (the tokens of) the prompt and the model response up to the answer location, and use `sae-lens` to generate only the next token. We cache the model activations at this step for RC.

# B   Prompts

12 prompt templates are used in our experiment. For each data point in a dataset, we sample 12 answers from the first prompt, 6 from the second, 4 from the third, 3 from the fourth, 2 from the fifth and sixth, and 1 from the rest. This way, we cover the need for responses for all prompt-sample configurations. For every dataset, the way of presenting the question is the same:

```
Question:  {QUESTION}
Candidate answers:
A: {ANSWER_A}
B: {ANSWER_B}
C: {ANSWER_C}
...
```

The prompts only slightly differ in their instructions:

Prompt 1:

```
You are a helpful AI assistant, answer the following question:
{QUESTION_AND_CANDIDATE_ANSWERS}
Think step by step.  Briefly justify your reasoning process, then put your
final chosen answer in the form:  [The answer is:  (X)] at the end.
```

Prompt 2:

```
You are a knowledgeable helper, look at the following question:
{QUESTION_AND_CANDIDATE_ANSWERS}
Let's break this question down step by step.  Write some short explanations
for your reasoning, then put your answer in the form:  [The answer is:
(X)] at the end of your response.
```

Prompt 3:

```
You are an expert in multiple choice questions, answer the following
question concisely:
{QUESTION_AND_CANDIDATE_ANSWERS}
Think about the question step by step.  Provide some brief explanations for
your thinking process.  Put your answer in the form:  [The answer is:  (X)]
to the end.
```

Prompt 4:

```
You are a helpful AI assistant, answer this question:
{QUESTION_AND_CANDIDATE_ANSWERS}
Think step by step about this question.  Add a brief justification for your
choice of answer.  Output your answer in the form:  [The answer is:  (X)]
at the end of your response.
```

Prompt 5:

```
Answer the following question:
{QUESTION_AND_CANDIDATE_ANSWERS}
Let's think step by step.  Provide short explanations of your thinking
steps.  At the end of your response, put your choice of answer in the form:
[The answer is:  (X)].
```

Prompt 6:

```
Here's a question I need you to help with:
{QUESTION_AND_CANDIDATE_ANSWERS}
Let's break down this question and think step by step.  Briefly outline
your reasoning process.  Output your choice of answer with the form:  [The
answer is:  (X)] to the end.
```

Prompt 7:

```
Look at the following question and answer it:
{QUESTION_AND_CANDIDATE_ANSWERS}
Think step by step.  List out your thinking.  Keep it short.  Put your
answer in the form:  [The answer is:  (X)] at the end of your response.
```

Prompt 8:

```
I have a multiple choice question which you are going to help with:
{QUESTION_AND_CANDIDATE_ANSWERS}
Let's think slowly and step by step.  First briefly output your thinking
process with short justifications, then finally output your answer in the
form:  [The answer is:  (X)].
```

Prompt 9:

```
Please help me answer the following question:
{QUESTION_AND_CANDIDATE_ANSWERS}
Look at the question step by step.  Explain your thoughts very briefly and
finally output the answer in the form:  [The answer is:  (X)].
```

Prompt 10:

```
Which candidate answer do you think is correct for this question:
{QUESTION_AND_CANDIDATE_ANSWERS}
Consider this question step by step with short explanations for your
```

```
thoughts, then put your answer in the form: [The answer is: (X)] at the
end of your response.
```

Prompt 11:

```
Here is a question in the multiple choice form with four potential answers:
{QUESTION_AND_CANDIDATE_ANSWERS}
Analyse the question and candidate answers with step-by-step thinking, then
state the correct answer in the form: [The answer is: (X)] at the end of
your outputs.
```

Prompt 12:

```
Below is a multiple choice question. Look at the question and the
candidate answers, select the correct one:
{QUESTION_AND_CANDIDATE_ANSWERS}
Think about it step by step, present short explanations for your thoughts.
At the end of your output, state your answer in the form: [The answer is:
(X)].
```

# C   Result Details

## C.1   Number of Points

For the task performance results in Section 4.1, we only report results for the test points where multiple answers exist among the responses. Table 6 shows the average number of points used for each experiment, and the percentage of such points among all test sets. The results are averaged over the specific prompt-sample configurations at each number of responses.

Table 6: Average number of test points (and percentage) where multiple answers exist among the responses. The number after dataset name indicates the total test points for that model.

| Model | Dataset | 12 responses | 6 responses |
|---|---|---|---|
| Llama3.1-8B-IT | CSQA (1200) | $508(42.40 \pm 2.57)\%$ | $401(33.43 \pm 1.29)\%$ |
|  | HSwag (3000) | $1602(53.42 \pm 0.83)\%$ | $1307(43.58 \pm 1.67)\%$ |
|  | MedMCQA (3000) | $1822(60.73 \pm 0.63)\%$ | $1524(50.8 \pm 0.42)\%$ |
|  | MMLU (3000) | $1295(43.17 \pm 1.07)\%$ | $1078(35.95 \pm 0.80)\%$ |
| Gemma2-2B-IT | CSQA (1200) | $730(60.89 \pm 1.86)\%$ | $582(48.50 \pm 1.91)\%$ |
|  | HSwag (3000) | $1994(66.50 \pm 4.02)\%$ | $1575(52.50 \pm 2.58)\%$ |
|  | MedMCQA (3000) | $2447(81.57 \pm 1.66)\%$ | $2131(71.06 \pm 1.82)\%$ |
|  | MMLU (3000) | $1932(64.43 \pm 2.90)\%$ | $1591(53.05 \pm 2.48)\%$ |
| Gemma2-9B-IT | CSQA (1200) | $576(48.02 \pm 2.95)\%$ | $464(38.67 \pm 2.52)\%$ |
|  | HSwag (3000) | $1178(39.27 \pm 1.53)\%$ | $938(31.27 \pm 0.81)\%$ |
|  | MedMCQA (3000) | $1813(60.44 \pm 1.63)\%$ | $1516(50.54 \pm 1.30)\%$ |
|  | MMLU (3000) | $1028(34.29 \pm 1.93)\%$ | $819(27.31 \pm 1.67)\%$ |
| Gemma2-27B-IT | CSQA (1000) | $405(40.50 \pm 2.08)\%$ | $309(30.90 \pm 1.58)\%$ |
|  | HSwag (1000) | $474(47.40 \pm 1.82)\%$ | $355(35.50 \pm 0.66)\%$ |
|  | MedMCQA (1000) | $505(50.50 \pm 0.61)\%$ | $4270(42.7 \pm 1.46)\%$ |
|  | MMLU (1000) | $429(42.90 \pm 0.81)\%$ | $345(34.50 \pm 0.76)\%$ |

We observe that it is common to obtain different answers from multiple responses of the same LLM, often more than 50% for each dataset. This is more obvious for smaller models as they might be less certain on their predictions. Also, it happens more frequently with more responses. Additionally, incorporating more prompt rephrases (e.g., comparing 6 prompts, 2 responses each with 2 prompts, 6 responses each) will result in more test points having different answers, contributing to the standard deviations

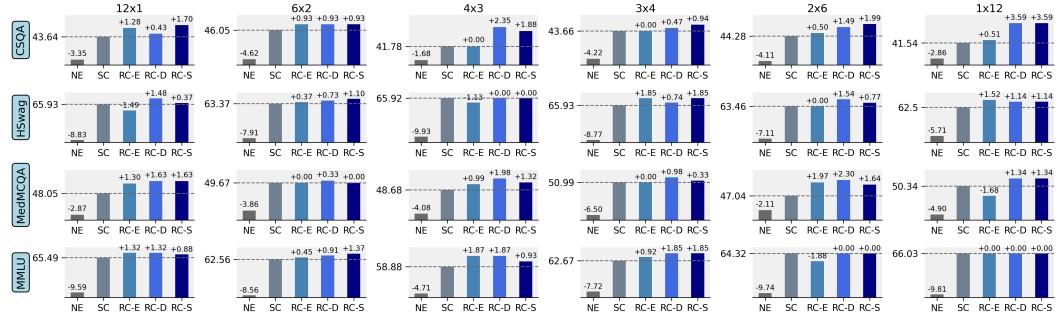

(a) Results for 12 responses.

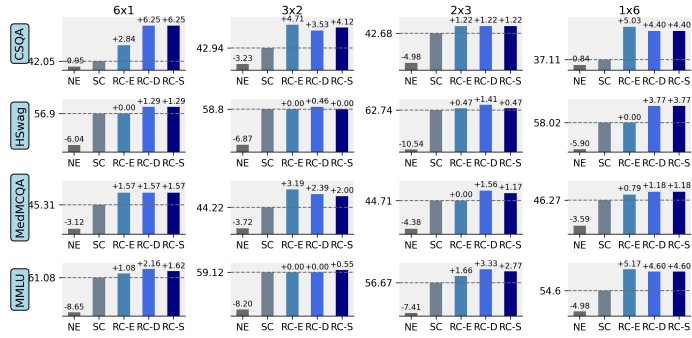

(b) Results for 6 responses.

Figure 3: All results for Llama3.1-8B-IT

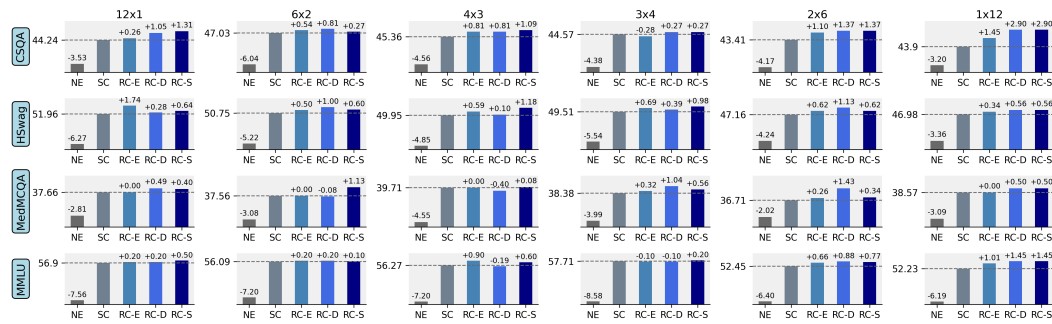

(a) Results for 12 responses.

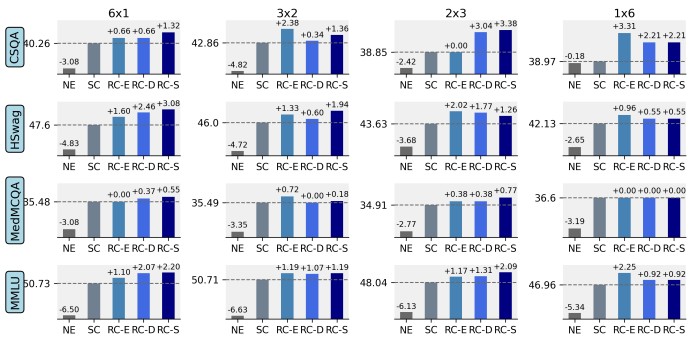

(b) Results for 6 responses.

Figure 4: All results for Gemma2-2B-IT

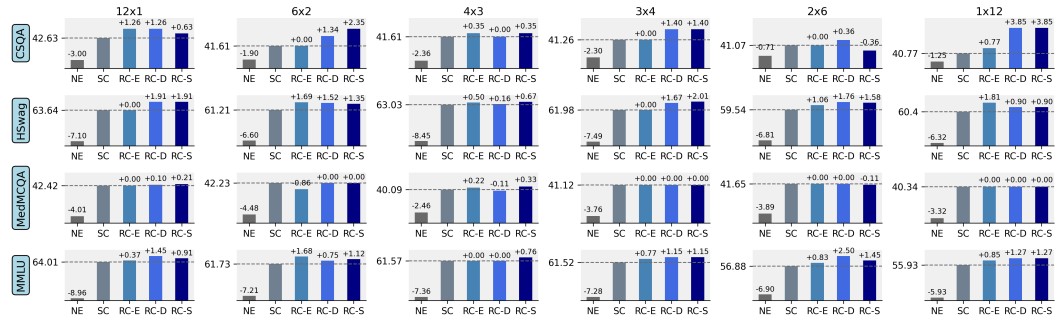

(a) Results for 12 responses.

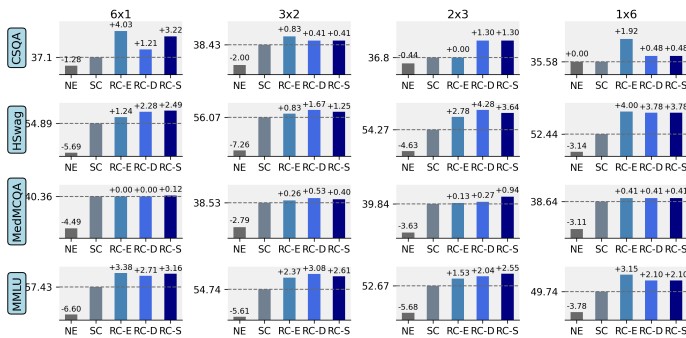

(b) Results for 6 responses.

Figure 5: All results for Gemma2-9B-IT

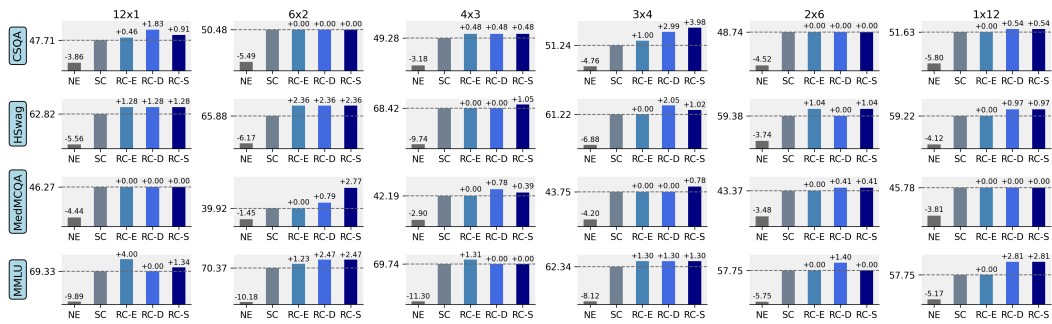

(a) Results for 12 responses.

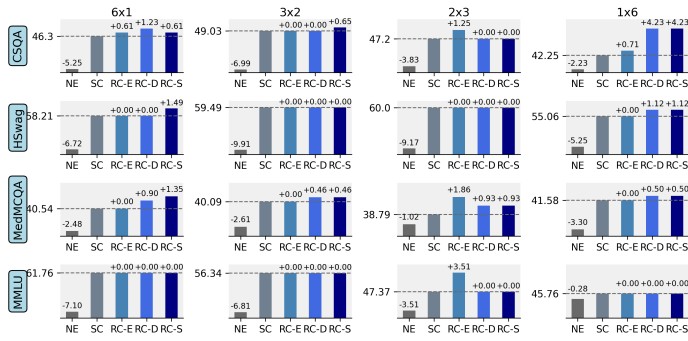

(b) Results for 6 responses.

Figure 6: All results for Gemma2-27B-IT

## C.2   Task Performance Results per Configuration

Figures 3 to 6 report the detailed results for each prompt-sample configuration in our experiments. The observations in Section 4.1 also apply to these results, although here we can observe a larger range of accuracy changes for RC-E, RC-D, and RC-S. For example, largest accuracy improvements for RC-D and RC-S are 6.25% for CSQA dataset on Llama model with 6 prompts and 1 sample per prompt. We can also see cases where the RC- methods worsen the accuracy from SC. This is because the optimal hyperparameters can be overfitted on the tuning subset of data, specifically if the answer distributions (e.g., the number of test points having 2 different answers, each with 6 supporting responses, versus the number of test points having 2 different answers with 2 and 10 supporting responses, respectively) are very different between the tuning subset and the test subset. This happens more frequently with the 27B model as there are fewer points in the test sets (Table 6).

## D   Ablation Analysis

We perform additional explorations on the RC method accuracy against the two hyperparameters in RC, namely $\lambda$ (ranging from -1 to 1), and LLM layer $l$ where the model activations are taken. While negative $\lambda$ values are not used in practice, we experiment with them to validate the usefulness of LLM activations. When $\lambda$ is negative, the evaluation function of RC (Equation 5) is calculated as: $\lambda \cdot consistency + (1 - (abs(\lambda))) \cdot frequency$.

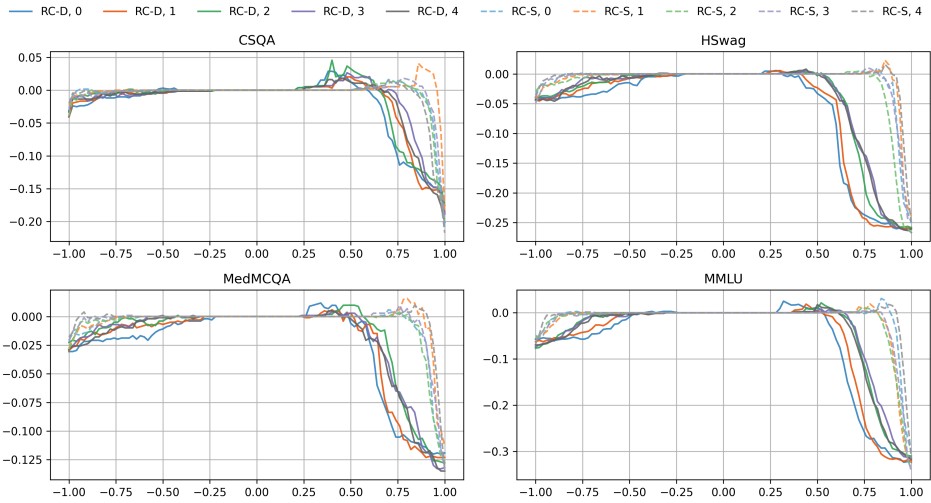

Figure 7: Ablation analysis on Llama3.1-8B-IT. The number following RC-D or RC-S indicates the index of layer $l$, see Table 5 for specific layer numbers.

Figures 7 to 10 report the relative accuracy difference (vertical axis) against varying $\lambda$ values (horizontal axis) for each model and each dataset, averaged over four 6-response configurations. The majority vote result (the SC baseline) is obtained at $\lambda = 0$. The only $\lambda$ value region with constant performance gain over the majority vote accuracy is when $0.25 \leq \lambda \leq 0.85$ (roughly, depending on dataset and the choice of dense or sparse activations). The performance steadily drops with small fluctuations as $\lambda$ becomes more negative.

For every line, when $\lambda$ takes values between about [-0.25, 0.25], the accuracy stays constant (with very little fluctuations) at the SC result (majority vote, $\lambda = 0$), because the frequency term is dominant in this region. When $\lambda$ increases from the above interval into the more positive region, the performance usually also starts to increase. The accuracy then quickly drops to a very low value because frequency no longer plays an important role (the method ends up choosing very infrequent answers). When $\lambda$ decreases from the above interval into the more negative region, in most cases, the performance will linearly decrease with fluctuations. At certain $\lambda$ locations, the performance can fluctuate to up to about 1% over the initial performance. RC-D reaches the performance peak and the performance drop at lower $\lambda$ values than RC-S.

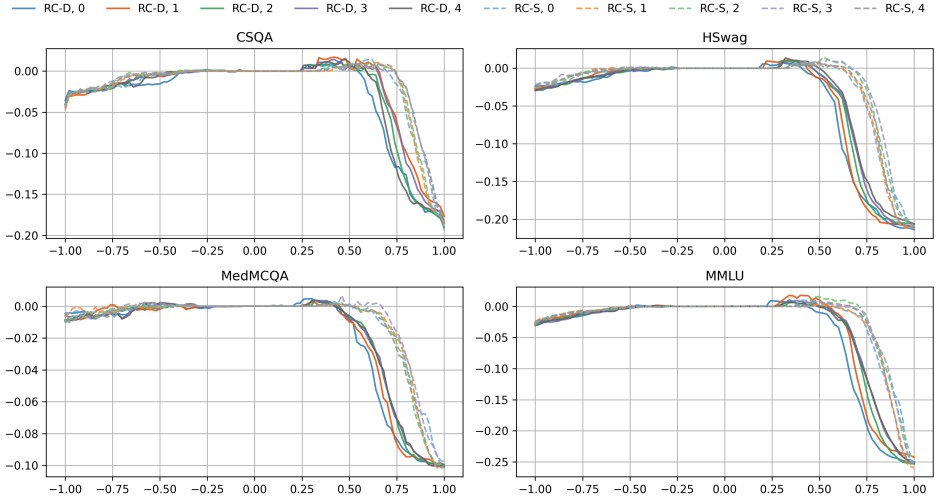

Figure 8: Ablation analysis on Gemma-2-2B-IT. The number following RC-D or RC-S indicates the index of layer $l$, see Table 5 for specific layer numbers.

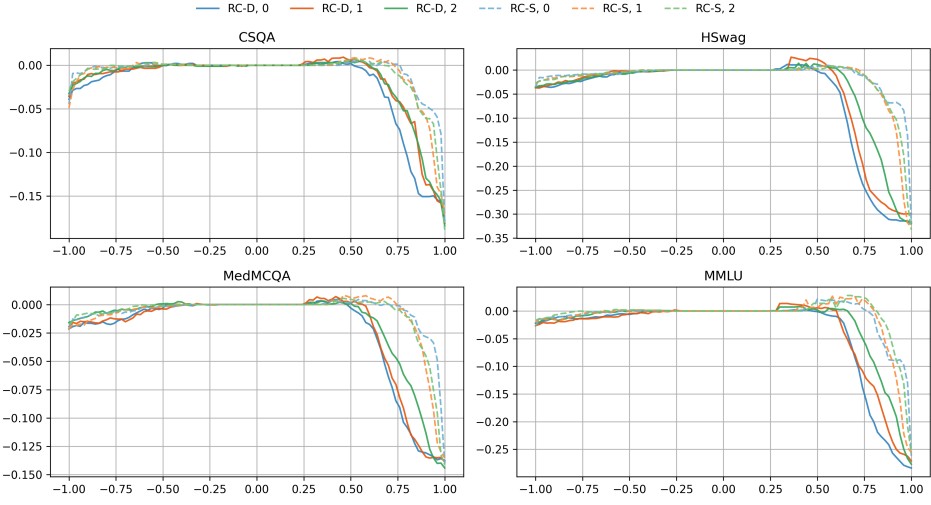

Figure 9: Ablation analysis on Gemma-2-9B-IT. The number following RC-D or RC-S indicates the index of layer $l$, see Table 5 for specific layer numbers.

The reason for a smaller performance drop at $\lambda = -1$ than when $\lambda = 1$ is that the method tends to choose the majority answer in the most negative $\lambda$ region. This is because when the difference between the number of supporting responses for multiple answers is large (which is true for most test points), the majority answer tends to have a lower consistency due to the involvement of more responses, therefore is more likely to be selected by the RC method. For example, for a data point we have 12 responses, 3 are predicting answer A and 9 are predicting answer B, and answer B usually has a lower consistency because. It will more likely be selected by $\lambda = -1$ than by $\lambda = 1$. This also highlights the importance of balancing consistency and frequency. We further note that very large absolute $\lambda$ values are impractical to use.

The differences between model layers $l$ are not particularly obvious, as similar performance gains can be observed for most configurations.

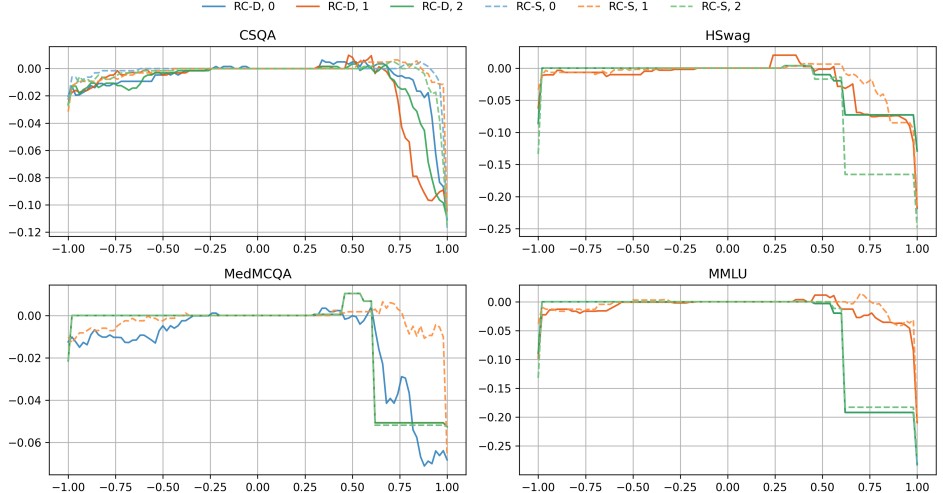

Figure 10: Ablation analysis on Gemma-2-27B-IT. The number following RC-D or RC-S indicates the index of layer $l$, see Table 5 for specific layer numbers.

# E  LLMs' Suitability to Evaluate Coherence

It should be noted that the results of Section 4.2 rely on the ability of LLMs to evaluate the concept of *coherence*. This reliance is supported by existing evidence: coherence is often used as a high-level evaluation metric during alignment training [70, 32], and LLMs are frequently employed as high-performing reward models [40]. To validate this capability in our specific context, we conducted a small-scale study. To validate their suitability in our context, we collected 10 examples from our experiments, each containing a question and 6 LLM responses evenly divided between two distinct answers (i.e., 3 responses per answer group). We manually labelled which group was more coherent and found that our LLM evaluator's judgements perfectly matched our labels on these data points. We caution, however, that assessing coherence may become more difficult in practice with larger response sets. Below, we present an illustrative example from the MMLU dataset. In this case, the responses predicting answer B all exhibit identical reasoning and are therefore considered more coherent than the group predicting answer D.

```
Question:  For their first exam, students in an AP Statistics class studied
an average of 4 hours with a standard deviation of 1 hour.  Almost everyone
did poorly on the exam, and so for the second exam every student studied
10 hours.  What is the correlation between the numbers of hours students
studied for each exam?
```

```
Candidate answers:  A: -1, B: 0, C: 0.4, D: 1
```

```
Correct answer:  B: 0
```

```
Group 1 responses predicting D:1
```

1. Here's how to think about this: **Correlation measures the strength and direction of a linear relationship between two variables.** **In this case, the two variables are the number of hours studied for the first exam and the number of hours studied for the second exam.** **Notice that every student studied 10 hours for the second exam, regardless of how much they studied for the first exam.** This means there's a perfect, direct relationship: the number of hours studied for the second exam is always 10, no matter what the first exam study time was. **A perfect, direct relationship has a correlation of 1.** [The answer is: D]

2. Here's how to determine the correlation:  **Understanding Correlation:** Correlation measures the strength and direction of the linear relationship between two variables. A correlation of -1 indicates a perfect negative linear relationship, 0 indicates no linear relationship,

and 1 indicates a perfect positive linear relationship. **Analyzing the Scenario:** In this case, we're looking at the relationship between hours studied for the first exam and hours studied for the second exam. **Key Observation:** Every student increased their study time by a fixed amount (9 hours) for the second exam. This means there's a perfect, direct relationship between the two study times. If a student studied 3 hours for the first exam, they studied 12 hours for the second. If they studied 5 hours for the first, they studied 14 hours for the second, and so on. **Conclusion:** Since there's a perfect, direct relationship, the correlation is 1. [The answer is: D]

3. Here's how to determine the correlation: **Understanding Correlation:** Correlation measures the strength and direction of the linear relationship between two variables. A correlation of -1 indicates a perfect negative linear relationship, 0 indicates no linear relationship, and 1 indicates a perfect positive linear relationship. **Analyzing the Situation:** In this case, every student increased their study time by a fixed amount (from an average of 4 hours to 10 hours). This means the relationship between study time for the first exam and the second exam is perfectly linear. **Perfect Linear Relationship:** Since the study time for the second exam is directly determined by the study time for the first exam (plus 6 hours), the correlation is 1. [The answer is: D]

```
Group 2 responses predicting B:0
```

1. Here's how to determine the correlation: **Understanding Correlation:** Correlation measures the strength and direction of the linear relationship between two variables. A correlation of -1 indicates a perfect negative linear relationship, 0 indicates no linear relationship, and 1 indicates a perfect positive linear relationship. **Analyzing the Situation:** In this case, we're looking at the number of hours studied for the first exam and the number of hours studied for the second exam. Since *all* students studied 10 hours for the second exam, regardless of how much they studied for the first exam, there is no linear relationship between the two. **The Answer:** [The answer is: B]

2. Here's how to reason through this: **Correlation measures the strength and direction of a linear relationship between two variables.** **In this case, we're looking at the relationship between hours studied for the first exam and hours studied for the second exam.** **The key information is that every student studied 10 hours for the second exam, regardless of how much they studied for the first.** This means there's no variation in study time for the second exam based on the first exam's study time. **Since there's no linear relationship (hours studied for the first exam doesn't predict hours studied for the second), the correlation is 0.** [The answer is: B]

3. Here's how to determine the correlation: **Understanding Correlation:** Correlation measures the strength and direction of a linear relationship between two variables. A correlation of -1 indicates a perfect negative linear relationship, 0 indicates no linear relationship, and 1 indicates a perfect positive linear relationship. **Analyzing the Situation:** In this case, we know that everyone increased their study time for the second exam. Since every student studied 10 hours for the second exam, regardless of how much they studied for the first exam, there is no linear relationship between the two. [The answer is: B]

# F  Impact Statements

This work focuses on using LLM model internals to aid answer aggregation from multiple responses, and can have important broader impacts. Practically, our positive accuracy improvement results (Section 4.1) suggest that the proposed method can be directly applied when open-source LLMs are used for short-form text generation tasks. In terms of research impact, our work bridges two traditionally separate research fields, test-time scaling (without model retraining) and mechanistic interpretability. While we do not propose new interpretability methods, the use of model internals is inspired by this line of research. As discussed in the Conclusion section, there are multiple directions for future research following this work. Our answer coherence results (Section 4.2) also motivate further research into transparency and interpretability of LLMs.

