# OpenReview forum: "Representation Consistency for Accurate and Coherent LLM Answer Aggregation"
_NeurIPS.cc/2025/Conference — NeurIPS 2025 poster_

### Official Review · Reviewer_mfsR · 2025-06-24

**Clarity:** 3
**Significance:** 2
**Originality:** 2
**Rating:** 3
**Confidence:** 3

**Summary:**

# Method summary
The authors look at the problem of inconsistency in LLMs, where LLMs are prone to generating different answers to similarly phrased questions. The authors propose modifying self-consistency (SC) to include "representation consistency", where the final consistency score is a linear combination of how _similar_ the representations across various outputs were and the frequency term which "measures the proportion of responses supporting each answer". In the sparse variant of this metric, the representations used in the first term are the representations extracted from sparse autoencoders. The method is directly applicable to QA like settings where it is easy to cluster the final answer but not a trivial extension to non-QA (non-discrete choices) settings.

# Experiment setup summary
The authors look at 4 QA datasets and the Llama and Gemma variants. The baseline considered across experiments is SC, i.e. omitting the first term of the authors' proposed method.

# Results summary
Table 1 shows ~1% absolute improvement over SC across the Llama and Gemma variants. Interestingly, Table 2 documents different values of optimal $\lambda$ for each model and type of RC. The authors finally then employ LLM-as-a-judge (Deepseek-R1) to judge coherence of the answer and report higher coherence for the sparse RC version.

**Questions:**

1. Can the authors clarify about Table 2 and provide a comprehensive table / chart and analysis on $\lambda$?
2. Does (and how does) this method hold in other scenarios like needle-in-a-haystack QA, adversarial reasoning, etc?

**Ethical Concerns:**

["NO or VERY MINOR ethics concerns only"]

**Final Justification:**

> I wanted to clarify by what I mean by improvements to the presentation:

> Experiment section: The larger figure can be replaced with more results and discussion around the results presented in the rebuttal
Extension to future / inclusion in current: When models choose to abstain or have "discussion with other LLMs" in their context window, how would the results change / what future works should look for (this requires additional experiments on the same datasets / newer datasets)
> For these reasons, I'm keeping my score.

**Limitations:**

The limitations are covered in the weaknesses and the lack of an answer to the second question in the paper is also a major limitation.

**Quality:**

2

**Strengths And Weaknesses:**

# Strengths

1. The proposed method is strong in the idea that LLMs are potentially better than the measured performance by looking at their generations and RC tries to expand on this by looking at their internal representations and hypothesize that representations could possibly be _more_ consistent than the generations.

# Weaknesses

1. The results and analysis around the results are weak. Table 2 for instance can explore a lot more on different depth variants and inform the readers on the variation of performance and optimal $\lambda$ values that could exist to better understand properties of these models (by model family and by number of parameters) - an incomplete analysis on Table 2 leaves me wondering how other layers perform.
2. LLM-as-a-judge with no apparent grounding leaves me wondering the validity of the results and the inference from it. The authors should have some baseline to check if the DeepSeek is capable of measuring coherence on a small subset of manually validated reasoning traces and only then scale to the experiment. This sets a bad precedent of "blindly" trust LLM-as-a-judge outputs without strong ground truth or a validation stage to ensure the judgement capability.
3. Key implementation details (number of runs, seeds if any, etc.) and confidence levels are missing in the individual charts to determine significance.

---

> ### Author Rebuttal · Authors · 2025-07-30
>
> We thank the reviewer for the insightful and constructive feedback. Below we provide our response to each question.
>
> ## **Response to Question 1 and Weakness 1 about parameters in our method:**
>
> We agree that more fine-grained sensitivity analysis will provide better insights into our proposed method. We therefore perform additional explorations on the effects of the two parameters, $\lambda$, and LLM layer $l$ where the model activations are taken. Note that the SAE latent dimension is fixed by the pretrained SAE models, therefore is not a parameter in our method.
>
> **Due to the restriction in uploading new results, here we describe our approach for this investigation and summarise the general findings. We plan to add this analysis as a section in the appendix and relevant discussions into Section 4.1 in the final version of the paper.**
>
> We create a 2-D line plot for each model and each dataset, averaged over four 6-response configurations (6 prompts x 1 sample), (3 x 2), (2 x 3), (1 x 6), yielding 16=4 models x 4 datasets plots. In each plot, the x-axis is the $\lambda$ value, ranging from -1 to 1 with an interval of 0.02, the y-axis is the accuracy. Each plot presents the accuracy of varying $\lambda$ for each ($l$ x {RC-D, RC-S}) combination. We experiment with negative $\lambda$ values to validate the usefulness of representation consistency. When $\lambda$ is negative, the evaluation function of RC (Equation 5) is calculated as: $\lambda\cdot consistency + (1-(abs(\lambda)))\cdot frequency$.
>
> The overall conclusion from the plots regarding $\lambda$ is: **The only $\lambda$ value region with constant performance gain over the majority vote accuracy is when $0.25 \leq\lambda \leq0.85$ (roughly, depending on dataset and the choice of dense or sparse activations). The performance steadily drops with small fluctuations as $\lambda$ becomes more negative.** Here are some more detailed observations:
>
> - For every line, when $\lambda$ takes values between about [-0.25, 0.25], the accuracy stays constant (with very little fluctuations) at the self-consistency result (majority vote, $\lambda=0$), because the frequency term is dominant in this region.
> - When $\lambda$ increases from the above interval into the more positive region, the performance usually also starts to increase for a $\lambda$ interval of about 0.1 with peak increase of about 4% (similar to Figure 2 (a), CSQA dataset with 6 responses). The accuracy then quickly drops (usually happens at $\lambda>0.85$) to a very low value of about 20-25% because frequency no longer plays an important role (the method ends up choosing very infrequent answers).
> - When $\lambda$ decreases from the above interval into the more negative region, in most cases, the performance will linearly decrease with fluctuations. The performance drop at $\lambda=-1$ is usually 5 to 10%. Meanwhile, at certain $\lambda$ locations, the performance can fluctuate to up to about 1% over the initial performance.
> - The reason for a smaller performance drop at $\lambda=-1$ than when $\lambda=1$ is that the method tends to choose the majority answer in the most negative $\lambda$ region. This is because when the difference between the number of supporting responses for multiple answers is large (which is true for most test points), the majority answer tends to have a lower consistency due to the involvement of more responses, therefore is more likely to be selected by the RC method. For example, for a data point we have 12 responses, 3 are predicting answer A and 9 are predicting answer B, and answer B usually has a lower consistency because. It will more likely be selected by $\lambda=-1$ than by $\lambda=1$. This also highlights the importance of balancing consistency and frequency. We further note that very large absolute $\lambda$ values are impractical to use.
> - The differences between model layers $l$ among the same choice of either dense or sparse are almost negligible.
> - RC-D reaches the performance peak and the performance drop at lower $\lambda$ values than RC-S (this is consistent with the findings in Table 2).
> - Because we are testing this on the whole dataset without the validation/test set split (different to our main experiments in the paper), RC-S almost always achieves higher accuracy than RC-D.
>
> These observations confirm the usefulness of representation consistency. Note that our Table 1 results are averaged for each model over four datasets and 10 prompt x sample configurations (4 for 6 responses and 6 for 12 responses). The Figure 2 results are slightly more fine-grained as they contain two plots for each model and each dataset, respectively averaged over the 4 configurations of 6 responses, and 6 configurations for 12 responses. The maximum accuracy gain in Figure 2 is 4% which is higher than those averaged in Table 1, and of course, there are cases where the performance gain is very small.
>
>
> ## **Response to Question 2 about method generalisability:**
>
> Theoretically yes, our formalisation for the method generalises to other supervised QA scenarios, as long as a concrete, relatively short-form answer can be parsed from the LLM response and can be straightforwardly aggregated across LLM responses, e.g., via semantic entropy [Kuhn, Gal, Farquhar, ICLR 2023]. It will require moderate efforts to generalise beyond multiple choice settings if we specify a list of possible answers for the question. It will require substantial efforts to generalise to more open-ended conversations, possibly involving an extra LLM as the parsing agent. We will mention these as future work.
>
> ## **Response to Weakness 2 about LLM-as-a-judge validity:**
>
> We thank the reviewer for pointing this out. We agree that validating LLMs' ability to perform relevant evaluation is an important point to address, which we did not mention in the initial paper. We had assumed the most recent powerful LLMs (including our chosen 671B DeepSeek R1) have learned the concept of "coherence" which we are evaluating. Despite the availability of internet corpus to train these LLMs, another argument for this assumption is that "coherence" is explicitly listed as one of five evaluation dimensions in two popular datasets for training reward models for LLM post-training - HelpSteer [Wang et al., NAACL 2024] and HelpSteer2 [Wang et al., NeurIPS 2024]. LLMs are often prompted as reward models and perform well, according to RewardBench leaderboard.
>
> More concretely, we have curated a tiny validation dataset of 5 data points for checking the LLM's performance on evaluating coherence. Similar to the approach described in Section 4.2, we look at test cases with six model responses where the model produces two answers, and each answer has three supporting model responses. These are taken from Gemma-2-9B-IT's model outputs on MMLU dataset. We manually label the group of answers which is more coherent, and compare the label generated by the LLM model via API call. The accuracy we observed here is 100%.
>
> We plan to include relevant discussions and a scaled-up version of the above validation experiment in the final version of the paper.
>
> ## **Response to Weakness 3 about implementation details:**
>
> Our setting assumes that the LLM answers have been generated. The remaining procedures give deterministic results. We did not test the method over multiple runs of LLM generations for the same data. The results in Figure 2 are averaged over the 4 and 6 configurations (prompt x sample) respectively for 6 and 12 responses. We report detailed results for every configuration in Appendix C.2.

---

> > ### Comment · Reviewer_mfsR · 2025-08-03
> > **Final thoughts**
> >
> > Thank you for the clear rebuttal. The methodology and experiments were sound, but I'd like to maintain my score since I strongly believe another revision in writing, presentation and a few more experiments on other QA datasets (math, reasoning, etc.) should make this a strong re-submission in a future venue.

---

> > > ### Author Response · Authors · 2025-08-07
> > > **Further rebuttal**
> > >
> > > Thank you again for recognising the contributions of our work and the soundness of the rebuttal. Your constructive feedback will make this paper better!
> > >
> > > Finally, we respectfully argue that integrating the additional material during rebuttal will not require substantial changes to the main text. The new content primarily serves to strengthen the soundness and completeness of our results, and most of it can be incorporated as appendices, without altering the current structure or claims of the submission. We believe they will further enhance the clarity and rigour of the paper.
> > >
> > > We also appreciate the suggestion to explore additional datasets and task types. Given the inherent structural differences between multiple-choice QA (our focus in this work) and other forms of QA generation, a thorough investigation would require non-trivial modifications. Therefore, we will mention these as promising directions for future work.

---

> > > > ### Comment · Reviewer_mfsR · 2025-08-08
> > > >
> > > > I wanted to clarify by what I mean by improvements to the presentation:
> > > >
> > > > 1. Experiment section: The larger figure can be replaced with more results and discussion around the results presented in the rebuttal
> > > > 2. Extension to future / inclusion in current: When models choose to abstain or have "discussion with other LLMs" in their context window, how would the results change / what future works should look for (this requires additional experiments on the same datasets / newer datasets)
> > > >
> > > > For these reasons, I'm keeping my score.

---

### Official Review · Reviewer_mQJA · 2025-06-27

**Clarity:** 4
**Significance:** 3
**Originality:** 3
**Rating:** 5
**Confidence:** 3

**Summary:**

This paper describes a method to increase the likelihood of choosing a correct answer from a pool of answers generated by the same/different LLMs. The main intuition is that we expect correct answers to be generated by similar activations in the latent space (implying higher model confidence), whereas incorrect answers might activate more diverse/dissimilar pathways (implying lower model confidence).

The authors test their hypothesis using 4 open-access models on 4 benchmarks Question-Answering datasets, and find the sparse variant of their approach typically outperforms the dense variant, as well as a strong baseline that simply chooses the modal answer from the pool.

**Questions:**

I might have missed it, but does RC work for a pool of answers generated by models of different sizes? For example, if you got answers from both Gemma-2B and Gemma-27B. My impression is that you can only vary the prompt for the same model, but I wondered if there were any plans to handle this case, and perhaps assign greater weighting to larger models which are more likely to be correct.

**Ethical Concerns:**

["NO or VERY MINOR ethics concerns only"]

**Final Justification:**

As stated above, I thought this was a well-motivated paper carried out with convincing experiments and results.
I did not feel that a detailed analysis of the sensitivity of the $\lambda$ and $l$ parameters raised by the other reviewers was crucial to the contribution of the paper, but welcome that addition nevertheless.

**Limitations:**

The authors note that their approach has a parameter that is both model- and dataset-specific, which limits its generalisability.

I would also suggest the authors acknowledge that it may not always be practical to generate >1 response per question (especially open-ended questions), as this invokes a larger time/computation cost, particularly for larger models.

**Quality:**

3

**Strengths And Weaknesses:**

The problem statement is well-described; Figure 1 is a fantastic representation of the idea!

The paper is generally clear, concise and convincing, and also well-structured.

The experiments and experimental setup are well-motivated and logical.

The authors make limitations of their work clear.

---

> ### Author Rebuttal · Authors · 2025-07-30
>
> We thank the reviewer for the positive feedback. Below we provide detailed responses to the comments.
>
> ## **Response to Question about different model sizes:**
>
> Our method works on various prompting and sampling strategies on the same model because we need to take its internal activations for different generation traces. While activations from different models will not be readily comparable, there is recent research on transferring information across the activations from different models [Activation Space Interventions Can Be Transferred Between Large Language Models, Oozeer et al., ICML 2025]. So this direction will be interesting future work.
>
> Additionally, in the setting of multiple models, one could employ our method in a composite answer aggregation framework where, an answer from each model is first determined by our method, then a higher-level aggregation is performed over each model's answer.
>
> ## **Response to Limitations**
>
> We appreciate the practical considerations raised by the reviewer. We agree that our targeted scenario is not the most generic, but including multiple prompts and samples can also be realistic, e.g., for prompt template selection and for uncertainty quantification. We will include this caveat in the limitation discussions in the final version of the paper.

---

> > ### Comment · Reviewer_mQJA · 2025-08-05
> >
> > I thank the authors for their response and clarification. I was unaware of the Oozeer et al. 2025 paper, and agree it would be interesting to adapt their method for different model sizes in future work.
> >
> > I also appreciate the authors acknowledgement of my observed limitation of their work, and that they will include it in the final version of the paper.
> >
> > Finally, having read the other reviews, I also agree that providing further information about the effects of changing the parameters $\lambda$ and $l$ strengthens their contribution, so welcome that discussion in the main paper.

---

> > > ### Author Response · Authors · 2025-08-07
> > >
> > > Thank you again for recognising the contributions of our work! We surely will include additional materials during the rebuttal to the final version of the paper!

---

### Official Review · Reviewer_8CA3 · 2025-07-03

**Clarity:** 3
**Significance:** 1
**Originality:** 2
**Rating:** 4
**Confidence:** 4

**Summary:**

The paper proposes a test-time scaling approach where you sample multiple times from the model then select an answer based on frequency sampled + avg cosine similarity of the internal representations. The motivation is that more consistent representations across different prompts may signal more accurate answers. Specifically, they're taking just the final embedding corresponding to the token where the reasoning model starts generating the final answer. They also do variants of this using an external embedding model or applying an SAE on top of the representations.

Results:
- Across 4 QA tasks, they evaluate Llama 8B instruct and Gemma-2 models and see that the performance improves by 0.5-1.8%.
- They use LLM judge to check the similarity of the model's chain of thought, and find that when the chains of thought are similar, the consistency (cosine similarity) of internal representations tend to also be high. But this is only true for the SAE representations.

**Questions:**

I think addressing the following questions might help better connect representation consistency with accuracy:
- See point above regarding CoT being a confounding factor. What if the model is forced to generate an answer with no CoT? Are the performance gains more significant there?
- If you measure the representation consistency of very frequent answers (say frequency threshold above $\tau$), is it indeed the case that incorrect answers have lower representation consistency?
- How much does representation consistency correlate with confidence? Is there something we're learning from representations that cannot be captured by answer confidence?
- Could you plot $\lambda$ on the x-axis and Accuracy on the y-axis and see how the performance interpolates for different lambda? - - - What if you considered negative $\lambda$ values? Is the performance strictly worse than Majority Vote?

**Ethical Concerns:**

["NO or VERY MINOR ethics concerns only"]

**Final Justification:**

I think the new experiments demonstrate that internal activations similarity might be providing information beyond output similarity. The experiments conducted during the rebuttal could be more robust (if I understood correctly, it looks like the statistics are measured over a small number of rollouts). But they show sufficient preliminary signs for the paper to be accepted.

**Limitations:**

There's a limitation paragraph on the conclusions.

**Quality:**

2

**Strengths And Weaknesses:**

While the idea of representation consistency is interesting, I think the paper could be improved upon in the following axes:
- The paper could be much stronger if it provided some more analysis that demonstrates that higher representation consistency truly does correspond with accuracy. (See questions below.)
- There seems to be a disconnect with the initial motivation of the paper, which was to make sure the representations are consistent across different prompts. The model is also allowed to do reasoning before generating the final answer, and the reasoning trace generated may also effect the representations + confound the results. For example, the representations might just be more similar because the reasoning traces are more similar, in which case, can you just measure how similar the reasoning traces are? I'm not sure what the representations are truly capturing in the experiments.
- The paper focuses on performance comparison, but the performance gains seem to be pretty small generally compared to majority vote (at most 1.8%). The gains do not seem strong enough to indicate that representation consistency is extracting something significant from the model.

---

> ### Author Rebuttal · Authors · 2025-07-30
>
> We thank the reviewer for the insightful feedback. We provide responses to the questions below.
>
> ## **Response to Question 1 and Weakness 2 about Non-CoT scenario:**
>
> Our target is on any combination of prompts and sampling strategies, e.g., 12 model generations from 2 prompts and 6 samples per prompt. This is more generic than "make sure the representations are consistent across different prompts". We decided to focus on CoT setting because this is a more general setting and allows the model to generate diverse reasoning traces, even for the same prompt.
>
> We might be measuring the similarity of reasoning trace, and that is an intended behaviour of the proposed approach. The model activation is taken at the final answering token position, i.e., after the CoT reasoning and before it decides which candidate answer to select. This representation captures the encodings for both the question and its own reasoning trace for answering the question, which is in line with our targeted scenario. This means that we are not just measuring the surface level syntactic similarity of reasoning (this is what the baseline, RC-E is doing), but the semantic similarity internal to the model's generation process.
>
> In terms of the non-CoT setting, we need to consider two types of configurations.
>
> - First, let's consider when multiple samples are needed. When we have one prompt only, the model activations for multiple samples are identical because this is taken at the end of: "... the final answer is:". The differences in the generated answering token between each sampled answer (e.g., A, B, C) only come from the decoding and the sampling process itself. Therefore, the consistency measure (Equation 6) will be 1 for any group of model outputs, and will be 1 for even outputs across different answers. The RC selection process (Equation 8) will be dominated by the frequency term. Our method therefore becomes identical to self-consistency. When we have multiple samples per prompt and additionally multiple prompts, it would not make sense to aggregate and calculate representation consistency, knowing that different answers might also have identical model activations. This is a major reason why we focus on CoT setting only.
>
> - Then, the only non-CoT configuration for RC to make sense is when we have multiple prompts and one sample per prompt. We did investigate applying RC to non-CoT outputs in early explorations, and the performance gains were similar to current results. However, we ignored these because such configurations (6 prompts x 1 samples, and 12 x 1) are only a fraction of our reported configurations.
>
> ## **Response to Questions 2, 4 and Weaknesses 1, 3 about performance gain and parameters:**
>
> We agree that more fine-grained sensitivity analysis will provide better insights into our proposed method. We therefore perform additional explorations on the effects of the two parameters, $\lambda$, and LLM layer $l$ where the model activations are taken.
>
> **Due to the restriction in uploading new results, here we describe our approach for this investigation and summarise the general findings. We plan to add this analysis as a section in the appendix and relevant discussions into Section 4.1 in the final version of the paper.**
>
> We create a 2-D line plot for each model and each dataset, averaged over four 6-response configurations (6 prompts x 1 sample), (3 x 2), (2 x 3), (1 x 6), yielding 16=4 models x 4 datasets plots. In each plot, the x-axis is the $\lambda$ value, ranging from -1 to 1 with an interval of 0.02, the y-axis is the accuracy. Each plot presents the accuracy of varying $\lambda$ for each ($l$ x {RC-D, RC-S}) combination. We experiment with negative $\lambda$ values to validate the usefulness of representation consistency. When $\lambda$ is negative, the evaluation function of RC (Equation 5) is calculated as: $\lambda\cdot consistency + (1-(abs(\lambda)))\cdot frequency$.
>
> The overall conclusion from the plots regarding $\lambda$ is: **The only $\lambda$ value region with constant performance gain over the majority vote accuracy is when $0.25 \leq\lambda \leq0.85$ (roughly, depending on dataset and the choice of dense or sparse activations). The performance steadily drops with small fluctuations as $\lambda$ becomes more negative.** Here are some more detailed observations:
>
> - For every line, when $\lambda$ takes values between about [-0.25, 0.25], the accuracy stays constant (with very little fluctuations) at the self-consistency result (majority vote, $\lambda=0$), because the frequency term is dominant in this region.
> - When $\lambda$ increases from the above interval into the more positive region, the performance usually also starts to increase for a $\lambda$ interval of about 0.1 with peak increase of about 4% (similar to Figure 2 (a), CSQA dataset with 6 responses). The accuracy then quickly drops (usually happens at $\lambda>0.85$) to a very low value of about 20-25% because frequency no longer plays an important role (the method ends up choosing very infrequent answers).
> - When $\lambda$ decreases from the above interval into the more negative region, in most cases, the performance will linearly decrease with fluctuations. The performance drop at $\lambda=-1$ is usually 5 to 10%. Meanwhile, at certain $\lambda$ locations, the performance can fluctuate to up to about 1% over the initial performance.
> - The reason for a smaller performance drop at $\lambda=-1$ than when $\lambda=1$ is that the method tends to choose the majority answer in the most negative $\lambda$ region. This is because when the difference between the number of supporting responses for multiple answers is large (which is true for most test points), the majority answer tends to have a lower consistency due to the involvement of more responses, therefore is more likely to be selected by the RC method. For example, for a data point we have 12 responses, 3 are predicting answer A and 9 are predicting answer B, and answer B usually has a lower consistency because. It will more likely be selected by $\lambda=-1$ than by $\lambda=1$. This also highlights the importance of balancing consistency and frequency. We further note that very large absolute $\lambda$ values are impractical to use.
> - The differences between model layers $l$ among the same choice of either dense or sparse are almost negligible.
> - RC-D reaches the performance peak and the performance drop at lower $\lambda$ values than RC-S (this is consistent with the findings in Table 2).
> - Because we are testing this on the whole dataset without the validation/test set split (different to our main experiments in the paper), RC-S almost always achieves higher accuracy than RC-D.
>
> These observations confirm the usefulness of representation consistency. Note that our Table 1 results are averaged for each model over four datasets and 10 prompt x sample configurations (4 for 6 responses and 6 for 12 responses). The Figure 2 results are slightly more fine-grained as they contain two plots for each model and each dataset, respectively averaged over the 4 configurations of 6 responses, and 6 configurations for 12 responses. The maximum accuracy gain in Figure 2 is 4% which is higher than those averaged in Table 1, and of course, there are cases where the performance gain is very small.
>
> Finally, as we note as a limitation in our conclusion (Lines 307-309), while incorporating representation consistency into answer selection enables performance gain at a dataset level, it does not guarantee a more truthful answer for every input. This is because the LLM might be confidently wrong about certain knowledge, where consistent reasoning (and activations) can lead to wrong answers.
>
> ## **Response to Question 3 about model confidence:**
>
> We assume the LLM "confidence" mentioned by the reviewer is referring to uncertainty quantification (UQ), which is itself an ongoing research question. The most obvious approach for this is to look at the token probability when the model is generating the final answers. In our early explorations, we observed over-confident token proabilities for different predicted answers across multiple responses for the same question. Recent research (reference [11] in our paper) uses some similarity metric based on LLM's dense internal activations over multiple model generations for UQ purpose and shows improved calibration (AUROC of using this new metric to predict LLM answers' correctness) over count-based UQ baselines (like Semantic Entropy [Kuhn, Gal, Farquhar, ICLR 2023]). This suggests that useful signals can be extracted from the LLMs' internal activation spaces for better UQ.
>
> Our method additionally considers the answer frequency, SAE sparsified signals, and multiple prompt x samples configuration, thus should intuitively correlate well with UQ and calibration. Indeed, the relationship between our method and LLM UQ could be interesting future work. We have mentioned this in our submitted paper at Lines 319-322.

---

> > ### Comment · Reviewer_8CA3 · 2025-08-06
> > **Thanks!**
> >
> > Sorry about the late response and thanks for your rebuttal!
> >
> > Great, so from this new experiment, it seems that representation consistency does give you nontrivial gains i.e. the best performance is when the regularization term is far above 0.
> > This address my concern about improvements being negligible.
> >
> > My second question isn't fully answered though. To reiterate:
> > _Consider all answers that are sampled frequently (i.e. by majority vote). Now divide these answers by if they are correct or incorrect. If the majority vote answer is incorrect, is it the case that the representation is less consistent than when the answer is correct?_
> >
> > Specifically, I wonder what the _distribution_ of similarities look like for these two groups of reasoning traces. Is the distribution unimodal or bimodal? What types of problems are we doing better on by adding the regularization term?
> >
> > Generally, I'm still feeling dubious about whether "internal representations of the model generating the answer" is really giving us useful information that **isn't present in the outputs**, specifically say that instead of internal representations, we used:
> > - Similarity metrics directly over  the reasoning trace:
> >     - For example, text embedding or n-gram similarity of the generations.
> >     - Is it just that model solutions tends to be more accurate if the reasoning trace is more similar?
> > - The model's average confidence of the model outputs
> >     - I think there was some confusion with what I meant by confidence in the original question.
> >
> > Going back through the submission + appendix, there isn't any analysis or experiment that attempts to answer this question. Overall, I still think there are many experiments to be completed to really support the submission's motivational claims/intuition about internal representations. Otherwise, it's not clear what the method is aiming to do at the core. There needs to be a thorough comparison with more baselines (confidence, similarity of reasoning traces, etc) and this makes me lean towards leaving the score as is.

---

> > > ### Author Response · Authors · 2025-08-07
> > > **Further rebuttal**
> > >
> > > Thank you for your response and, again, for your insightful comments! We are glad that the rebuttal has addressed some of your concerns.
> > >
> > > ### **Part 1**
> > >
> > > For the remaining points, we’d like to first answer the question about additional information gained from model activations that isn’t present in the outputs. Note that in the paper, we included a baseline RC-E, which works in a similar way to our proposed RC methods but calculates the consistency term directly using similarities of text embedding over the reasoning traces (model generations). So in that sense, we already have the baseline with reasoning trace.
> > >
> > > Regarding model confidence, we were thinking about uncertainty quantification methods as measures for model confidence. We have considered using token probabilities as model confidence, but have observed overconfidence in predicted tokens at the answering locations. Furthermore, aggregated token probability is not a reliable measure of confidence over a long model response since it quickly shrinks to near-zero values. The self-consistency (taking majority vote over multiple traces) method can also be seen as a confidence-quantifying (uncertainty quantification) method, as discussed in their original paper. This view has also been reiterated later in the Semantic Entropy paper and reference [15] in our submission – measuring occurrences of the same answer across samples and different semantically equivalent prompts helps decompose sources of uncertainty (aleatoric and epistemic), leading to more reliable confidence measure. Therefore, in this sense, we already have a baseline using model confidence over each answer.
> > >
> > > Our experiment results therefore show that better accuracy could be achieved with the help of internal activations than with semantic similarity and with model confidence measures. Moreover, the RC method combines frequency and activation consistency, so it actually includes confidence as part of the method.
> > >
> > > Another intuition why internal activations could carry more information than the output texts, beyond existing evidence in mechanistic interpretability literature, is as follows: suppose there are multiple (e.g., three) valid reasoning traces that lead to the correct answer for a given question. Due to the stochastic nature of the sampling process, any single LLM generation may only reflect one of those reasoning paths in the output text. However, the internal activations may still encode information necessary for the LLM to produce the other valid reasoning traces, even if they are not explicitly realised in the current generation. This suggests that the model’s internal state may represent a richer set of reasoning possibilities than what is surface-visible in any single output.
> > >
> > > ### **Part 2**
> > >
> > > About your Question 2 in the review: we collected the test cases where among the 12 responses, 6 support one answer and the other 6 support another answer, and one of these two answers is the correct one. We then count the number of times the correct vs. incorrect answer is associated with a higher consistency (dense activation consistency (ours), sparsified activation consistency (ours), and embedding consistency (baseline)). We aggregate the results over all prompt x sample configurations and over the datasets for each model. This is presented in the table below:
> > >
> > > | Method/Model | Gemma-2-2B-Inst (497) | Gemma-2-9B-Inst (404) | Gemma-2-27B-Inst (113) | Llama-3.1-8B-Inst (195) |
> > > |----------|----------|----------|----------|----------|
> > > |Textual embedding (baseline)| 50.9% vs. 49.1%   | 49.5% vs. 50.5%   | 47.8% vs. 52.2%   | 50.3% vs. 49.7%   |
> > > |Dense activations (ours)| 53.3% vs. 46.7%   | 53.7% vs. 46.3%  | 57.5% vs. 42.5%   | 56.4% vs. 43.6%   |
> > > |Sparse activations (ours)| 53.9% vs. 46.1%    | 54.7% vs. 45.3%   | 58.4% vs. 41.6%   | 56.4% vs. 43.6%   |
> > >
> > > *The number after each model name is the number of interested cases matching the description. The two numbers in each entry are percentages where the correct vs. incorrect answer is associated with a higher consistency.*
> > >
> > > **We observe that the correct answer more frequently demonstrates a higher consistency calculated with internal activations, than with textual embeddings.** We originally did not perform this experiment because we thought that the RC method (and the baseline RC-E or similar ensembling method) is prone to the cases where the LLM is confidently wrong about some answer, so *such measures could be tightly linked to the LLM’s performance*. This is also discussed in our limitations in the conclusion section.
> > >
> > > But now we see that this new experiment could confirm the non-trivial improvement with internal activations, and that they do help reveal information during the response generation process that is not included in the textual outputs.
> > >
> > > We will include relevant discussions in the main text (where space allows) and this new experiment as an appendix in the final paper. We hope that these make our contribution clearer!

---

### Official Review · Reviewer_pyGd · 2025-07-03

**Clarity:** 4
**Significance:** 3
**Originality:** 3
**Rating:** 4
**Confidence:** 4

**Summary:**

The paper proposes Representation Consistency (RC), a test-time answer reranking method that considers not only the frequency of each candidate answer but also the consistency of internal representations (activations) of the language model across different responses that lead to the same answer. Responses are grouped by their final predicted answer, and a consistency score is computed for each group based on pairwise similarity of their hidden states. The final answer is selected via a weighted combination of frequency and representation consistency. The authors also explore a sparse variant (RC-S) that leverages pretrained Sparse Auto-Encoders (SAEs) to improve the interpretability and robustness of the representation space. Experiments on four reasoning benchmarks and multiple open-source LLMs demonstrate consistent gains over self-consistency (SC), a strong test-time scaling baseline.

**Questions:**

No

**Ethical Concerns:**

["NO or VERY MINOR ethics concerns only"]

**Final Justification:**

I’ve read the rebuttal and updated my score. This paper demonstrates some unique strengths.

**Limitations:**

See Weaknesses

**Quality:**

2

**Strengths And Weaknesses:**

**Strengths:**

   - The method is simple, effective, and well-motivated.
   - RC captures an important intuition: correct answers tend to be supported by semantically coherent reasoning paths.
   - The sparse variant (RC-S) is particularly compelling in aligning with human notions of coherence.
   - The paper is clearly written and experimentally thorough.



**Weaknesses:**

   1. **Comparison with MBR missing:** While not a direct instantiation of Minimum Bayes Risk (MBR) decoding, RC shares similar motivations. It would be useful to compare RC with classical MBR-style answer selection methods, especially since both aim to aggregate over multiple candidate outputs [1].
   2. **Limited sensitivity analysis of hyperparameters:** The method depends on several hyperparameters (e.g., λ for weighting, the activation layer l, and the SAE latent dimension d_SAE). Although optimal values are reported, the paper lacks a thorough sensitivity analysis on how performance varies with these parameters.
   3. **Lack of evaluation on mathematical reasoning tasks and other generative tasks:** RC is not tested on dedicated math benchmarks like GSM8K, which are standard for evaluating test-time scaling in reasoning. Since SC was originally proposed for such tasks, it would be valuable to understand how RC performs in this domain. Besides, I recommend that the authors include experiments on generative tasks (e.g., NQ and TriviaQA).
   4. **Computational cost of RC-S not reported:** RC-S relies on pretrained Sparse Auto-Encoders (SAEs), which may introduce significant overhead. The paper does not discuss the cost or feasibility of training these SAEs or reusing them across tasks.
   5. **Limited analysis on response set size:** The method is evaluated with 6 and 12 responses, but a more detailed analysis across varying numbers of responses (e.g., 2, 4, 8) would help assess the robustness and scaling behavior of RC versus SC.



[1] Epsilon sampling rocks: Investigating sampling strategies for minimum Bayes risk decoding for machine translation.

[2] Ensemble Learning for Heterogeneous Large Language Models with Deep Parallel Collaboration.

[3] Bridging the Gap between Different Vocabularies for LLM Ensemble

---

> ### Author Rebuttal · Authors · 2025-07-30
>
> We thank the reviewer for the helpful and insightful comments. Below, we provide detailed responses to each question.
>
> ## **Response to Question 1. Comparison with MBR missing:**
>
> Our adapted answer selection baseline, RC-E, builds on the same intuition as MBR decoding. It is described in Lines 171-190 in the paper's main text - taking the final answer which is frequent and whose Chain-of-Thought outputs are the most semantically similar. Our methods, on the other hand, focus on the similarity in each candidate LLM output's internal representation and outperform this baseline.
>
> Additionally, both the MBR method and the reference [1] listed by the reviewer are orthogonal to our proposed method. Our method operates on multiple LLM outputs regardless of the sampling strategies with which they are obtained. Therefore, investigating the influence of different decoding process on the performance of RC would be an interesting future work, but is out of the scope of this study. We mentioned this line of future work in the paper's main text at Lines 314-316.
>
> The other two references [2] and [3] listed by the reviewer target different scenarios from ours. They resolve the conflicts between different vocabularies across multiple LLMs, whereas our method works on multiple generations for the same LLM with a focus on the internal activations. References [1] [2] [3] all focus strongly on machine translation, which is also different to our setting. We will add pointers to these references as related works in the final version of our paper.
>
>
> ## **Response to Question 2. Limited sensitivity analysis of method parameters:**
>
>
> We agree that more fine-grained sensitivity analysis will provide better insights into our proposed method. We therefore perform additional explorations on the effects of the two parameters, $\lambda$, and LLM layer $l$ where the model activations are taken. Note that the SAE latent dimension is fixed by the pretrained SAE models, therefore is not a parameter in our method.
>
> **Due to the restriction in uploading new results, here we describe our approach for this investigation and summarise the general findings. We plan to add this analysis as a section in the appendix and relevant discussions into Section 4.1 in the final version of the paper.**
>
> We create a 2-D line plot for each model and each dataset, averaged over four 6-response configurations (6 prompts x 1 sample), (3 x 2), (2 x 3), (1 x 6), yielding 16=4 models x 4 datasets plots. In each plot, the x-axis is the $\lambda$ value, ranging from -1 to 1 with an interval of 0.02, the y-axis is the accuracy. Each plot presents the accuracy of varying $\lambda$ for each ($l$ x {RC-D, RC-S}) combination. We experiment with negative $\lambda$ values to validate the usefulness of representation consistency. When $\lambda$ is negative, the evaluation function of RC (Equation 5) is calculated as: $\lambda\cdot consistency + (1-(abs(\lambda)))\cdot frequency$.
>
> The overall conclusion from the plots regarding $\lambda$ is: **The only $\lambda$ value region with constant performance gain over the majority vote accuracy is when $0.25 \leq\lambda \leq0.85$ (roughly, depending on dataset and the choice of dense or sparse activations). The performance steadily drops with small fluctuations as $\lambda$ becomes more negative.** Here are some more detailed observations:
>
> - For every line, when $\lambda$ takes values between about [-0.25, 0.25], the accuracy stays constant (with very little fluctuations) at the self-consistency result (majority vote, $\lambda=0$), because the frequency term is dominant in this region.
> - When $\lambda$ increases from the above interval into the more positive region, the performance usually also starts to increase for a $\lambda$ interval of about 0.1 with peak increase of about 4% (similar to Figure 2 (a), CSQA dataset with 6 responses). The accuracy then quickly drops (usually happens at $\lambda>0.85$) to a very low value of about 20-25% because frequency no longer plays an important role (the method ends up choosing very infrequent answers).
> - When $\lambda$ decreases from the above interval into the more negative region, in most cases, the performance will linearly decrease with fluctuations. The performance drop at $\lambda=-1$ is usually 5 to 10%. Meanwhile, at certain $\lambda$ locations, the performance can fluctuate to up to about 1% over the initial performance.
> - The reason for a smaller performance drop at $\lambda=-1$ than when $\lambda=1$ is that the method tends to choose the majority answer in the most negative $\lambda$ region. This is because when the difference between the number of supporting responses for multiple answers is large (which is true for most test points), the majority answer tends to have a lower consistency due to the involvement of more responses, therefore is more likely to be selected by the RC method. For example, for a data point we have 12 responses, 3 are predicting answer A and 9 are predicting answer B, and answer B usually has a lower consistency because. It will more likely be selected by $\lambda=-1$ than by $\lambda=1$. This also highlights the importance of balancing consistency and frequency. We further note that very large absolute $\lambda$ values are impractical to use.
> - The differences between model layers $l$ among the same choice of either dense or sparse are almost negligible.
> - RC-D reaches the performance peak and the performance drop at lower $\lambda$ values than RC-S (this is consistent with the findings in Table 2).
> - Because we are testing this on the whole dataset without the validation/test set split (different to our main experiments in the paper), RC-S almost always achieves higher accuracy than RC-D.
>
> These observations confirm the usefulness of representation consistency. Note that our Table 1 results are averaged for each model over four datasets and 10 prompt x sample configurations (4 for 6 responses and 6 for 12 responses). The Figure 2 results are slightly more fine-grained as they contain two plots for each model and each dataset, respectively averaged over the 4 configurations of 6 responses, and 6 configurations for 12 responses. The maximum accuracy gain in Figure 2 is 4% which is higher than those averaged in Table 1, and of course, there are cases where the performance gain is very small.
>
> ## **Response to Question 3. Lack of evaluation on mathematical reasoning tasks and other generative tasks:**
>
> We appreciate the reviewer's suggestion to include more benchmarks. Our goal in this paper is to evaluate RC's performance on general reasoning tasks with multiple-choice output format, for which we believe our choice of four datasets is sufficient. The chosen datasets are also widely adopted in prior works.
>
> In terms of adapting our method to other forms of generation tasks, this will require non-trivial extensions to be able to process and aggregate the possible discrete answer choices among the LLM generations, possibly with the help of semantic entropy [Kuhn, Gal, Farquhar, ICLR 2023] for shorter form answers (supervised QA) or another LLM judge for longer form open-ended conversations. Our compute resources are shared by a large group and so are limited. We will mention these suggestions as future work.
>
> ## **Response to Question 4. Computational cost of RC-S not reported:**
>
> In Lines 149 and 191-195, we discussed that the SAEs we used are pretrained models. We do not claim the training of such SAEs as part of our contribution. The SAEs are generally reusable, and RC-S only requires negligible additional computational cost compared with RC-D.
>
> **Reusability of SAEs:** SAEs are usually trained for one LLM at a time and then can be generally reused for this LLM, because the SAE training uses the LLM's activations over the LLM's own pretraining data (see Section 3.1 of reference [40]). In fact, these SAEs are widely used in the mechanistic interpretability literature to investigate LLM behaviours related to their internal activations with sparsified signals. Training SAEs is an active research field with public benchmarks (e.g., SAEBench: A Comprehensive Benchmark for Sparse Autoencoders in Language Model Interpretability, ICML 2025). The Python library SAE-Lens contains a list of supported SAEs for various LLMs. We took the open-weight SAEs in this library for our experiments.
>
> **Additional computational costs of SAEs:** If the layer number and lambda value are determined, the additional cost of this over RC-D for an input text is one forward pass in the SAE. Specifically, the input to SAE, LLM's residual activation at one token position, is of shape (1, $d_\pi$), where $d_\pi$ is the LLM's hidden dimension (2304 for Gemma-2-2B, 3584 for 9B, 4608 for 27B, 4096 for Llama-3.1-8B). The number of parameters in each SAE is: 0.076B for Gemma-2-2B, 0.117B for Gemma-2-9B, 1.2B for Gemma-2-27B, 0.27B for Llama-3.1-8B model. The small SAE model size (compared to their corresponding LLM) and the small input size (only 1 token length) mean that the additional computational cost is only a fraction of the LLM's forward pass over a prompt.
>
> We will add relevant discussions to the main text of the paper, should it be accepted.
>
> ## **Response to Question 5. Limited analysis on response set size:**
>
> Please note that in each number of responses configuration, we further have different Prompt x Samples configurations. So, in 6 responses we included (6 prompts x 1 sample), (3 x 2), (2 x 3), (1 x 6); in 12 responses we included (12 prompts x 1 sample), (6 x 2), (4 x 3), (3 x 4), (2 x 6), (1 x 12). These make 10 different experiment settings for each model and dataset, which we considered sufficient.

---

> > ### Comment · Reviewer_pyGd · 2025-08-06
> >
> > Thanks for your rebuttal, which addresses my concerns (1, 2, 4, and 5).
> >
> > In particular, I now recognize the essential difference: (1) MBR selects the candidate answer that is most similar to other answers, which works well for generative tasks; and (2) RC selects the candidate answer with the highest representational self-agreement, which mainly targets multiple-choice tasks.
> >
> > I will increase my score and hope these comments are helpful for improving the paper.
> >
> > However, my concern (3) — regarding how RC can be applied to generative tasks — still remains. I encourage the authors to explore this direction further, as addressing this challenge could significantly enhance the quality and impact of the paper.

---

> > > ### Author Response · Authors · 2025-08-07
> > >
> > > Thank you for recognising that the rebuttal has addressed most of your concerns. Your constructive feedback will surely improve our paper!
> > >
> > > We also appreciate the suggestion to explore additional datasets and task types. Given the inherent structural differences between multiple-choice QA (our focus in this work) and other forms of QA generation, a thorough investigation would require non-trivial modifications. Therefore, we will mention these as promising directions for future work.

---

### Note · Authors · 2025-08-13

We sincerely thank the reviewers and the area chair for a thorough evaluation.

Our proposed method, using LLM internal activations during generation to aid answer selection from multiple LLM responses, is based on the common observation that LLM activations could capture more information than the surface-level text outputs, and the intuition that if the model’s representations of multiple responses converging on the same answer are highly variable, this answer is more likely to be the result of incoherent reasoning.

We appreciate that all reviewers praised the effectiveness (pyGd, mQJA, mfsR) and interestingness (8CA3) of the idea and the method, and confirmed the paper's clarity. Reviewers pyGd, mQJA also emphasised that the experimental sections are "thorough", "well-motivated and logical".

We thank the reviewers for acknowledging that most of their questions have been addressed. Specifically, 3 reviewers (pyGd, 8CA3, mfsR) raised similar concerns regarding the need for a more thorough ablation study of our method’s parameters. In response, we conducted additional experiments, which show a steady performance increase over the majority vote (the self-consistency baseline) results as the weighting term for model activation consistency increases (up to a moderate value). Further, following reviewer 8CA3's constructive suggestion, we performed another experiment showing that the correct answer more frequently demonstrates a higher consistency calculated with internal activations, than with textual embeddings over surface-level text outputs. These additional analyses further support the significance of our results.

Reviewers pyGd and mfsR suggested exploring question-answering formats beyond multiple-choice tasks. While we agree this is an interesting direction, we believe our current results provide strong evidence for the effectiveness of our method in our targeted multiple-choice setting. In the submission, we evaluate on four widely used benchmarks, each using 10 configurations: (12 prompts x 1 sample each prompt), (6×2), (4×3), (3×4), (2×6), (1×12), (6×1), (3×2), (2×3), and (1×6). Our results indicate that our methods consistently outperform the baselines. Given the non-trivial additional effort required for a thorough study of other QA formats, we feel this would be better pursued as future work.

We value this constructive rebuttal process and will integrate the relevant discussions and new results into the final version of our work.

---

### Decision · Program_Chairs · 2025-09-17

**Decision:**

Accept (poster)

**Comment:**

This paper proposes an answer reranking method that, instead looking at consistency of the final answers, considers the internal representations of responses that lead to the same answer, and picks the one with the highest consistency. The paper demonstrates that this improves over simple self-consistency.

The reviewers say the method is simple, effective and the paper is clearly written. Various concerns were raised initially, many of which the authors address, and most of which the reviewers appreciate (see below).

A concern I must mention is that the gains do not appear strong; the authors rebuttal is that the method achieves a peak gain of 4\%. I would hesitate to over-index on this result.

However, I recommend acceptance since this approach is simple and shows promise and can be a stepping stone to new ideas.


Below are notes from the reviewers and the paper.
----

- The authors provided the sensitivity analysis that reviewers `pyGd`,  `8CA3` ask for (who acknowledged it although `mQJA` suggests this was not necessary)
- `pyGd` raised various important concerns (response set size is too limited, computational cost) which the authors addressed convincingly.
- `8CA3` brought up an excellent point about the analytical side of the paper: it doesn't quite show that the internal representations truly say something new about the correctness, on top of what the response itself does. **The authors address this with a new experiment on textual embeddings. Please incorporate this in the main paper.**
- Some reviewers say the gains seem weak (`8CA3, mfsR`). **The authors say that max accuracy gain is 4\% which seems to have convinced the reviewers -- as mentioned above, I take this with a pinch of salt.**
- Some reviewers requested going beyond MCQ datasets (`pyGd`, `mfsR `) -- the authors argue that this is a well-established set of datasets in this line of work; besides other datasets would require significant extensions for aggregation. This convinced `pyGd`s but not `mfSR`; I believe this is acceptable defense.
-  `mfSR` also questioned the use of LLM-as-a-judge and righly points out the undesirable precedent this may set.  The authors provide an experiment cross-verifying that the LLM-as-a-judge is a sensible metric. **Regardless, I'd mention this as a caveat to the results.**
- `pyGd` pointed out a missing comparison -- authors say this is orthogonal, which the reviewer agreed to.  Please make a mention of this in the paper for the sake of the readers.

From further discussion with `8CA3` (not present in the review), they pointed out some papers on reasoning exploring notions of internal representation & consistency that would be worth citing. Quoting `8CA3`: For example, [1] uses a latent decoder to predict the answer from different intermediate layers and selects the most consistent answer, and [2] uses dissimilarity of internal representations to sample diverse generations. The effect may be similar to those that choose answers that remain consistent across diverse prompts [3, 4].
- [1] https://arxiv.org/abs/2405.18711
- [2] https://arxiv.org/abs/2505.16552
- [3] https://arxiv.org/abs/2503.02670
- [4] https://arxiv.org/abs/2502.15924